# GNAQPMS v1.1: Accelerating the Global Nested Air Quality Prediction Modeling System (GNAQPMS) on Intel Xeon Phi Processors

Hui Wang[1,2,4,*], Huansheng Chen[2,*], Qizhong Wu[1,4], Junmin Lin[3], Xueshun Chen[2], Xinwei Xie[3], Rongrong Wang[1,4], Xiao Tang[2], Zifa Wang[2]

[1]College of Global Change and Earth System Science, Beijing Normal University, Beijing 100875, China
[2]LAPC, Institute of Atmospheric Physics, Chinese Academy of Sciences, Beijing 100875, China
[3]Intel (China) Corporation, Beijing 100013, China
[4]Joint Centre for Global Changes Studies, Beijing Normal University, Beijing 100875, China
*These authors contributed equally to this work and should be considered co-first authors

Correspondence to: zifawang@mail.iap.ac.cn

**Abstract.** The Global Nested Air Quality Prediction Modeling System (GNAQPMS) is the global version of the Nested Air Quality Prediction Modeling System (NAQPMS), which is a multi-scale chemical transport model used for air quality forecast and atmospheric environmental research. In this study, we present the porting and optimization of GNAQPMS on a second-generation Intel Xeon Phi processor, codenamed "Knights Landing" (KNL). Compared with the first-generation Xeon Phi coprocessor (codenamed Knights Corner, KNC), KNL has many new hardware features such as a bootable processor, high-performance in-package memory and ISA compatibility with Intel Xeon processors. In particular, we describe the five optimisations we applied to the key modules of GNAQPMS, including the CBM-Z gas-phase chemistry, advection, convection and wet deposition modules. These optimisations work well on both the KNL 7250 processor and the Intel Xeon E5-2697 V4 processor. They include 1) updating the pure Message Passing Interface (MPI) parallel mode to the hybrid parallel mode with MPI and OpenMP in the emission, advection, convection and gas-phase chemistry modules; 2) fully employing the 512-bit-wide vector processing units (VPUs) on the KNL platform; 3) reducing unnecessary memory access to improve cache efficiency; 4) reducing the thread local storage (TLS) in the CBM-Z gas-phase chemistry module to improve its OpenMP performance; and 5) changing the global communication from writing/reading interface files to MPI functions to improve the performance and the parallel scalability. These optimisations greatly improved the GNAQPMS performance. The same optimisations also work well for the Intel Xeon Broadwell processor, specifically E5-2697 v4. Compared with the baseline version of GNAQPMS, the optimised version was $3.51\times$ faster on KNL and $2.77\times$ faster on the CPU. Moreover, the optimised version ran at 26% lower average power on KNL than on the CPU. With the combined performance and energy improvement, the KNL platform was 37.5% more efficient on power consumption compared with the CPU platform. The optimisations also enabled much further parallel scalability on both the CPU cluster and the KNL cluster scaled to 40 CPU nodes and 30 KNL nodes, with a parallel efficiency of 70.4% and 42.2%, respectively.

## 1 Introduction

Insatiable computing demand is driven by the ever-increasing scientific demands in many research codes such as the climate model, Community Earth System Model (CESM), and the weather model, Weather Research and Forecasting Model (WRF). In the early days of computing, when there was insufficient computation capability, scientists had to make trade-offs to fit the computation into a limited budget. One example is that the physical, chemical and dynamic processes in the models were simplified to adapt to the limited computation capability. Another example is that the horizontal and vertical resolutions were also sacrificed owing to the limited computation capability. This means that many details were neglected or simplified in the model and that the simulation ability of the models was limited.

Until early 2000, the application performance could easily be increased by using higher frequency processors. As the semiconductor manufacturing technology improved, we reached the power density and thermal limitation of silicon technology in early 2000 for single-core processor design. The industry has taken a "right-hand turn" to deliver performance through more compute cores rather than by increasing processor frequency. As a result, applications need to embrace parallelism to achieve higher performance. At the same time, heterogeneous computing is widely used in the scientific computing area. Typical examples of many-core architecture include the graphics processing unit (GPU) and the Intel Many Integrated Core (Intel MIC) [Chrysos G, 2012].

With the popularity of the new architecture, geo-scientific models have been partially or fully ported to the GPU and MIC heterogeneous computation platforms to get better computation performance. There are many reports about porting models to the GPU heterogeneous platform. The Princeton Ocean Model (POM) (Xu et al., 2015), except the initialisation and input/output (I/O) modules, was fully ported to the GPU by using CUDA-C. Moreover, the model computation performance was improved on both single node and clusters. For the atmospheric chemistry models, the RADM2 chemical scheme in the WRF-CHEM model (Grell et al., 2005) was ported to different multi-core platforms. Although the application exits the limitation of the on-chip memory, the GPU version still gets a speedup of 8.5× when compared with its serial version (Linford et al., 2009). Similar to the GPU, the first-generation Intel Xeon Phi coprocessor (codenamed Knights Corner or KNC) was connected to the mainboard via the Peripheral Component Interface Express (PCI-E) bus (Xu et al., 2015), and the bandwidth of PCI-E has become the new performance bottleneck for some memory bandwidth-bounded software applications, e.g. the popular atmospheric model WRF on KNC (Meadows, 2012). Mielikainen et al. (2014a,b,c,2015a,b) did a series of works to transplant the physical schemes to the KNC platform in WRF, including the Goddard microphysics scheme, the Thompson microphysics scheme, the Goddard shortwave radiation scheme and the advection scheme in the model dynamic core. Among these works, the Goddard microphysics scheme (Tao and Simpson, 1993; Khain et al., 2003) got a 4.7× speedup on KNC and a 2.8× speedup on the CPU compared with its baseline version, and sharing the same modern hardware features led to a speedup on both the MIC and the CPU platform. In addition, this phenomenon of performance improvement also appeared in the optimisation work of Thompson cloud microphysics. In our work, the global atmospheric chemistry model GNAQPMS also got a speedup on both the CPU and the MIC platform after the optimisation.

As emphasised by Mielikainen et al. (2014a), making full use of the new hardware features of chips is the key to getting a performance improvement on the MIC platform. KNL is the second-generation Intel MIC architecture processor (Sodani, 2015). Compared with the CPU, KNL has more cores, 16 GB of on-chip Multi-Channel Dynamic Random Access Memory (MCDRAM), wider vector register and AVX-512 instructions support, and other minor architectural features. Compared with the GPU, KNL is a bootable processor and can work alone without a host CPU, which eliminates the bottleneck of the PCI-E bandwidth. In addition, KNL has adopted the x86 architecture and shares the same programming model as the Intel processors. This study focused on the optimisation of GNAQPMS to fully utilise the features provided by modern (and future) processors. These optimisations not only improve the performance of GNAQPMS on the KNL platform, but also work with our current and future generation of processors, e.g. Skylake. GNAQPMS was designed for global atmospheric aerosol and chemistry simulation. Its applications include the temporal and spatial evolution of atmospheric composition, providing the boundary conditions for regional models, intercontinental long-range transport and long-term climate change, and it also acts as a key component of the Earth System Model of the Chinese Academy of Sciences (CAS-ESM). Currently, GNAQPMS can only run on the CPU platform, and its parallel scalability and computation speed are about eight CPU nodes and 46 h per model year, respectively, at 1°×1° resolution (excluding the model I/O). This model's performance is suitable for short-term or medium-term (5 years or less) simulation, but not for long-term simulation (30 years or more). Moreover, the overhead of the model will further increase when the model I/O is included or a

higher model resolution (e.g. 0.25°×0.25°) is used. Therefore, the model cannot be directly coupled to an earth system model and used for long-term climate change simulation. By optimisation of the model codes and usage of new hardware, we aimed to greatly improve the model's parallel scalability and computation speed. The target computation speed in the future is about 5-10 model years per day (including the model I/O) at 0.25°×0.25° resolution. We plan to improve the parallel computation speed of GNAQPMS in the first step and improve the model I/O efficiency in the next step. The optimisation methods in this paper are also suitable for other atmospheric chemistry transport models that use similar chemistry or physical schemes to those of GNAQPMS. In general, the optimisation processes include three steps: 1) testing the baseline version codes and searching for the performance bottleneck; 2) discovering and applying the optimisation solutions according to the specific performance bottleneck; 3) and testing the codes and validating the new version codes. The optimisation process is iterative; that is, these steps would be repeated until the peak or satisfactory performance is reached. The single-node performance should be optimised prior to the multi-node optimisation. More details about the common ways to modernise the codes can be found at Intel's website (https://software.intel.com/en-us/modern-code/training/short-video-series).

The rest of the paper is organised as follows. Sec. 2 introduces GNAQPMS and the KNL processor. Sec. 3 presents the optimisation processes for GNAQPMS. Sec. 3.1 shows the methods and tools used for testing the baseline codes and finding the bottlenecks, followed by subsections describing the optimisation measures in detail. The numerical experiments of the performance testing are presented in Sec. 4, which includes the result validations in Sec. 4.2 and the performance tests in Secs. 4.3 and 4.4. The conclusions are given in Sec. 5.

## 2 Model and KNL description

GNAQPMS is a global multi-scale chemical transport model developed by the Institute of Atmospheric Physics, Chinese Academy of Sciences (Chen et al, 2015). The baseline version works on the x86 CPU platform. As far as we know, this is the first work that ported GNAQPMS to and optimises it for the KNL platform. The model description of GNAQPMS and KNL is presented as follows.

### 2.1 Model description of GNAQPMS

GNAQPMS is the global version of NAQPMS (Chen et al, 2015; Wang et al., 2006). NAQPMS is a 3D regional Eulerian model that has been widely applied to simulate the chemical evolution and transport of ozone (Li et al., 2007; Tang et al., 2010), aerosol and acid rain over East Asia (Wang et al., 2002; Li et al., 2011; Li et al., 2012) and to provide routine air quality forecasts in mega cities such as Beijing, Shanghai and Guangzhou (Wang et al., 2010; Wu et al., 2012; Wang et al., 2009). The typical time scale of these applications is several days or months. Unlike the application of NAQPMS, that of GNAQPMS usually has a typical time scale of more than 10 years and has very high requirements to model the computation speed (e.g. 5-10 model years per day). Therefore, we chose GNAQPMS to start the code optimisation. Figure 1 shows the framework of GNAQPMS; its model inputs include the meteorology field and static emissions, and its physical/chemical processes include dynamic emissions with profile assigned, advection, diffusion and convection due to the meteorology field, and gas chemistry, aerosol module, mercury chemistry and dry/wet deposition processes. GNAQPMS has several key techniques, including process analysis and tracer-tagging techniques, which will help to assess the contribution to emission sources (Wu et al., 2011). It is also a multi-scale nested and parallel computation model, and can be coupled to a regional model to simulate the air pollution from a global scale to a regional scale, and even to a city scale, with MPI functions on a high-performance parallel computation platform. The air pollutant concentration, depositions and source apportionment results will be outputted after the simulation.

As mentioned above, the key chemical processes in GNAQPMS contain gas-phase chemistry, aqueous phase chemistry and aerosol chemistry. The gas-phase chemical module is the CBM-Z mechanism (Zaveri and Peters, 1999), with the solver module updated by Feng et al. (2015) by using a modified backward Euler (MBE) method. In this study, the CBM-Z module was optimised heavily, as it is one of the most time-consuming modules in GNAQPMS, as shown in Figure 2. Other chemical reaction modules such as the aqueous-phase and the aerosol chemical module are relatively minor time-consuming modules compared with the CBM-Z module. The wet deposition module and the aqueous-phase chemical module use the RADM2 mechanism (Ge et al., 2014; Chang et al., 1987; Wang et al., 2002); the former is also a hotspot in GNAQPMS, which gets a good performance after being optimised. The other physical processes in GNAQPMS include dry deposition (Wesely, 2007), advection (Walcek and Aleksic, 1998; Walcek, 2000), diffusion and convection, and all of these modules are also important hotspots.

**2.2 KNL description**

The benchmark many-core processor we used in this study was the Intel Xeon Phi KNL 7250 processor. Compared with the first-generation KNC MIC coprocessor, KNL has many improvements. Similar to the GPU, KNC is a coprocessor and it cannot work alone. KNC needs a host CPU and must be connected to the mainboard via the PCI-E interface, and the bandwidth of PCI-E should be taken into account when designing codes for KNC. Unlike KNC, KNL can work alone as a processor like a normal CPU, which means more efficient memory access. Moreover, KNL is equipped with 16 GB of MCDRAM, whose bandwidth is higher than that of normal DDR4 yet lower than that of the on-chip caches. MCDRAM was designed to bridge the bandwidth gap between DDR4 and on-chip cache. The MCDRAM on KNL can be configured in three modes for different applications, namely, cache mode, flat mode and hybrid mode. Since GNAQPMS is not limited by the memory working set size detected by the VTune Memory-Access tool, the cache mode was chosen in our experiment. The core number and clock speed in KNL were also improved. The core number was increased from 61 to 68, and the frequency of each core was increased from about 1.2 GHz to 1.4 GHz at the same time. More details about KNL can be found at Intel's website (https://www.intel.com/content/www/us/en/products/processors/xeon-phi/xeon-phi-processors.html).

**3 Optimisation technology**

In this study, some optimisation measures were used when porting GNAQPMS to the KNL platform, including updating the pure MPI to a hybrid parallel mode, strengthening of the vectorisation, reducing unnecessary memory access, reducing the thread local storage (TLS) and changing the way the global communication works in GNAQPMS. Moreover, the baseline version and the optimised version of GNAQPMS were labelled as "Base-V" and "Opt-V", respectively.

**3.1 Baseline performance test**

The first step of the optimisation was to test Base-V GNAQPMS and to identify the hotspots of the model. As shown in Figure 2, the runtime breakdown of each section of Base-V GNAQPMS was measured and calculated by the MPI function mpi_wtime in the experiment on the x86 CPU platform. The top five time-consuming sections were the CBM-Z chemistry, wet deposition, advection, diffusion and emission modules. To analyse the insight performance bottleneck of GNAQPMS, we used the Intel VTune Amplifier (https://software.intel.com/en-us/intel-vtune-amplifier-xe/), Intel Advisor (https://software.intel.com/en-us/intel-advisor-xe) and Intel Trace Analyzer and Collector (ITAC; https://software.intel.com/en-us/intel-trace-analyzer). The VTune tools can do the analysis of the performance in high-performance computing (HPC), memory access, thread profiling with locks and waits analysis, floating-point operations per second (FLOPS) and floating point unit (FPU) utilisation analysis and detection of

hotspot functions. By using the VTune HPC performance detection tool, we could report the general performance, e.g. GFLOPS, bandwidth, CPU and FPU utilisation, through a simple report. Table 1 presents the general indicators detected by the VTune HPC performance detection tool for the two models. Moreover, an obvious increase in GFLOPS was detected from 93.741 to 279.479. The Memory Bound in Table 1 indicates the fraction of slots where a pipeline could be stalled owing to the demand load or store instruction, and the values of 9.2% of Base-V and 12.7% of Opt-V indicate that the bandwidth is not the limitation of our model. The FPU utilisation was also improved from 2.9% to 9.6%, although there is still room for improvement. Further analysis of the hotspots and bandwidth could be detected by Hotspot and Memory-Access in VTune, respectively. Hotspots are the segment codes that consume most of the time during the running of the model. Therefore, optimising these hotspot parts will be more helpful to improve the speed and efficiency of the model codes, and Figure S1 in the supplement shows the hotspots in Base-V GNAQPMS with low CPU utilisation. Moreover, using the Intel Advisor tool could help to learn about the vectorisation and bandwidth situation of the hotspot functions and modules. The Intel Advisor tool could also provide some information about the limitation of vectorisation or the reasons why vectorisation cannot be realised, as well as the primary solutions to the users to do the full vectorisation work. Furthermore, the realisation of multi-threads could be done with the help of the Intel Advisor tool. As for the MPI performance, ITAC could provide the parallel MPI balance information and communication profiling, which is auto-visualised by ITAC to analyse the MPI performance, as shown in Figure S2. However, the more significant step is designing the corresponding solutions for the hotspots or bottlenecks with the help of the tools mentioned above. Moreover, timely test and validation should be done after the optimisation. This whole process, as mentioned in Sec. 1, could be repeated many times to try different alternatives and gain a satisfactory performance.

Our optimisation is based on the tools and processes mentioned above. To achieve the goal of porting GNAQPMS from the CPU platform to the KNL platform, we adopted the basic idea of fully using the hardware features of KNL, e.g. multiple hyper-threads, vector computing units, MCDRAM and multi-level caches. Accordingly, the main optimisation technologies include changing the parallel mode, fully vectorising the codes and improving the cache hit rates. Base-V GNAQPMS uses only the MPI parallel mode, which would ignore the hyper-threads of the CPU as well as of KNL and may greatly limit the scalability owing to the expensive communication as the number of processes increases. Because of the knowledge limitation of the original designer of GNAQPMS, the old way (i.e. file reading and writing) of doing the global communication was used in Base-V GNAQPMS, which directly reduces the speed and limits the scalability.

### 3.2 Main optimisation methods

According to the profiling and analysis of Base-V GNAQPMS on the CPU platform, the following optimisation measures were conducted: 1) altering the pure MPI parallel mode to hybrid parallel mode with MPI and OpenMP functions; 2) manual strengthening of the vectorisation with the help of compiler directives to fully use the vector computation on the KNL platform; 3) reducing unnecessary memory access to improve the utilisation efficiency of caches; 4) reducing the TLS for the common variables of each OpenMP thread; and 5) changing the way global communication works from writing/reading interface files to using MPI functions. KNL contains many low-frequency cores, and each core contains four hyper-threads. Considering the relatively expensive overhead of communication in MPI for many cores of KNL, we adopted the hybrid parallel mode by using OpenMP and MPI. Furthermore, the OpenMP threads could fully use the hyper-threads in KNL, and the cheap communication cost of OpenMP could help to improve the scalability. As shown in Table 2, the optimisation measures and the corresponding speedup for the relatively high overhead sections are presented, and the optimisation steps in the heading of Table 2 refer to the optimisation measures mentioned in the preceding paragraph. OpenMP was added to the sections including the emission calculation, advection and convection, diffusion, gas-phase chemistry and wet deposition modules. The other sections did not adopt the OpenMP

optimisation because of the relatively low calculation density and time consumption, and the cost of establishing and destroying threads in these sections is larger than the benefits gained from OpenMP. Therefore, the use of OpenMP may lead to a decrease in performance for these modules. To ensure that the peak performance of OpenMP was fully achieved, we removed the TLS of the common variables in the CBM-Z module, which is effective in reducing the overhead of establishing the threads by reducing the

procedure for copying the common variables for each thread. The TLS is introduced for variables in named common blocks when using the threadprivate OpenMP directive and is allocated for each thread on thread creation. These variables are private for each thread but are global within the thread. When a thread references a common variable in its TLS, the memory address of the TLS is first located by calling an OpenMP library function with the thread ID, and then the common variable is addressed within the TLS space. Even for the references to common variables within the same named common block in the same subroutine, the above

process is repeated for every variable, rather than addressing the TLS and common block only once. Since calling the OpenMP library function for TLS addressing is expensive, and there are many references to these common variables in the user subroutines, the total overhead of using common variables in the TLS is extremely high. Linking against the static OpenMP library can partially alleviate the calling cost, but the cost is still prohibitive.

For global communications, an improvement was achieved by changing how communication is taking place. The original way for

global communication is by writing the messages that need to be broadcast to other processes to files, and then the processes need to read these files to get the message through an I/O channel, which is a bottleneck in the model. The old way has relatively low efficiency and will impact the performance greatly, especially in the initialisation module. Multiple processes that read the same file will make this file a critical section, and the limitation of the I/O bandwidth would also slow the speed. This problem was introduced owing to the lack of consideration for parallel computation in the early development of GNAQPMS. Instead of

writing/reading interface files, the new way is to use the MPI_ALLREDUCE and MPI_GATHERV functions to perform the global communications.

Manual strengthening of the vectorisation with compiler directives is used in the sections including emission, advection, diffusion and CBM-Z gas-phase chemistry. KNL supports 512-bit vector operations and data path. It consists of two VPUs that can perform up to two 512-bit vector operations per cycle. A previous study about the optimisation of the physical schemes in WRF

(Mielikainen et al., 2014a) included plenty of ways to vectorise the code and align the data for vectorisation, which was prepared for the upcoming unified AVX-512 instruction set on KNL and Skylake architecture CPUs. Although the compiler can automatically vectorise loops with no obvious data dependence, there are still many loops that cannot be optimised automatically because of loop or array dependencies. Therefore, manually adding vectorisation directives and reconstructing the loops are needed with the help of the Intel Advisor tool mentioned above. During this process, different optimisation tips were used for various

scenes, and typical vectorisation techniques were introduced in Secs. 3.4 and 3.5. As reported by Mielikainen et al. (2014a), alignment directives were added in the codes for ensuring that vectorisation results in peak performance. For KNL, if the data are aligned and padded to 64-byte boundaries, the efficiency of data access can be improved and vector operation can be executed with high efficiency. This operation is treated as part of the vectorisation optimisation and is isolated as an independent optimisation measure.

Memory optimisation is also a critical spot that should be concentrated on. As mentioned in Sec. 2.2, the MCDRAM on the KNL platform can be configured in three modes. Since the internal memory is not the bottleneck for GNAQPMS, 16 GB of MCDRAM is used as the last level cache for GNAQPMS. To utilise the two level caches and MCDRAM well, we had to remove some unnecessary memory access operations via optimisation and cut off some temporary arrays. In the original code, some array variables are allocated, used and de-allocated many times in the outermost loop of the time step. In the optimised code, these

variables are allocated and de-allocated only once outside the time-step loop. Moreover, reforming the loop order to realise vectorisation also enables the cache hit rates to be improved at the same time.

The optimisation details for the typical physical and chemical modules, including the initialisation, emission, advection, convection, diffusion, chemistry and deposition modules in GNAQPMS, are presented in the following sections.

### 3.2.1 Global communication

The global communication of the model parameter in Base-V GNAQPMS is realised through writing and reading interface files in MPI parallel computing. GNAQPMS does many global communications when the model is initialised for the defined model domains, grids and boundaries setting and such model parameter. Thus, the model initialisation gets a good speedup through this optimisation method, which can save the time consumed by the input/output resources. According to the performance experiment shown in Table 2, the speedup for this section reached 1.27 on the CPU and 1.26 on KNL compared with the Base-V model on the CPU platform. KNL has more processors than the CPU and will involve more MPI tasks and more communications between each task; thus, KNL gets a limited benefit through this measure on a single node. However, this optimisation method greatly improves the model scalability in the multi-node testing. The scalability test results are shown in Sec. 4.4. However, we should clarify that this optimisation measure is only suitable for GNAQPMS since this problem was caused by the lack of concern for parallel efficiency by the original designer of the model. Because of the low frequency of the KNL cores, the I/O writing/reading way may lower the performance of KNL more than that of the CPU, which is indicated by the single-node test in Sec. 4.3.

### 3.2.2 Emission process section and typical vectorisation

In GNAQPMS, the emission process section would read the external emission files and assign them to emission variables, and increase the relevant pollution concentration when the model is running. In a word, the emission process section prepares the emission data for GNAQPMS. Therefore, it is the first section in the calculation loop of one time step. The emission section calculates and distributes the emission rates of the relevant species for each vertical and horizontal layer and completes the unit conversion.

In our study, strengthening of the vectorisation by adding directives and constructing loops and multithreading were done in the emission section. The sample code of this work is shown in Figure 3 to explain the whole processes as an example. First, we changed the nesting order of loops from **j**, **i**, **igas** to **igas, j, i,** to ensure that the data would be continuously accessed and to improve the efficiency of the caches. Second, since the subroutine get_ratio_emit() is too big to be inlined automatically by the Intel compiler, we manually inlined it in the calling site of the main program to improve the calling efficiency and facilitate the vectorisation. Third, vectorisation was involved in the emission section in the model by using the parameters to convert a scalar structure of assignment value to variables to a vector structure, which is shown in step 3 in Figure 3. Finally, we added the directives, clauses, declaration and syntax comment of OpenMP outside the outermost loop, as shown in box "4". According to the performance testing of the sample code of this hotspot, it could get an 8.57× speedup on the CPU (E5-2697 V4) with two OpenMP threads. However, in the actual application, the number of OpenMP threads should fit the whole application to get the peak performance. This type of optimisation is common in Opt-V GNAQPMS. With the doubly wider vector registers in KNL and OpenMP optimisation, the speedup of the whole emission section reached 12.83× (from 167.09 s to 13.02 s).

Moreover, the allocatable arrays loading the emission rates of all species were kept, which had been de-allocated at the end of the emission module in Base-V GNAQPMS, since they would be used again in the gas-phase chemistry section in the same way. Therefore, the cost of allocating, assigning and de-allocating these arrays for the second time in the section of CBM-Z is eliminated

by preserving the variables across functions. Finally, the initialisation of these arrays was also updated from one statement to assign the whole four-dimension arrays to loops with OpenMP to initialise the values, thus improving the efficiency of the initialisation.

### 3.2.3 CBM-Z gas-phase chemistry section

The gas-phase chemistry module is the key module in GNAQPMS; this module uses the CBM-Z scheme (Zaveri and Peters, 1999; Chen et al., 2015). According to the performance analysis with the MPI timing function shown in Figure 2, the CBM-Z module is one of the most important and sophisticated hotspots in GNAQPMS.

The framework of the CBM-Z module is shown in Figure 4. It contains many complicated subroutines to calculate the gas-phase species concentration. The analysis of the algorithms and code structure is in the first place before the optimisation, and the flowcharts of the module are presented in Figure 4. Deep analysis with the Intel VTune tools showed that the most complicated and important hotspot was the IntegrateChemistry subroutine, which contains the subroutines SelectGasRegimes, PeroxyRateConstants, GasRateConstants, Setgasindices, MapGasSpecies and ODEsolver. The function of the SelectGasRegime subroutine is to choose the optimum combination of gas-phase chemistry mechanisms on the basis of the concentrations and emissions of different gas species. The selection of different gas-phase chemistry mechanisms controls the following progresses in the IntegrateChemistry subroutine. The subroutines PeroxyConstants and GasRateConstants calculate the gas reaction rates for the selected chemistry mechanisms by relying on the result of the SelectGasRegimes subroutine. Then, the following SetGasIndices subroutine prepares the index of the local concentration and emission variables, and the MapGasSpecies subroutine converts the global gas species concentration variables to the local concentration variables when it is called for the first time. After that, the MBEsolver subroutine would calculate the ordinary differential equation functions of the gas-phase chemistry reactions. The second call to the MapGasSpecies subroutine returns the new values of the global gas concentrations.

According to the code structure of the CBM-Z module, the optimisation work was done step by step. At first, because of the relatively simple structures and functions, the Setrunparameters and SolarzenithAngle subroutines, as shown in red in Figure 4, were removed and made into inline functions to improve the calling efficiency. For the subroutines in yellow in the CBM-Z module, including the PrintResult and IntegrateChemistry subroutines, strengthening of the vectorisation was conducted. The PrintResult subroutine has a function for converting the units of gas concentration from molecules/cc to ppb with one loop, and a pragma directive was added to this loop to force the compiler to do the vectorisation.

In the CBM-Z module, the core calculation is in the IntegrateChemistry subroutine, whose flowchart is also shown in the right plot in Figure 4. The optimisation of this subroutine contributed the most remarkable performance improvement for GNAQPMS. The main optimisation of the IntegrateChemistry subroutine includes two parts of work: strengthening of the vectorisation by constructing a vector loop and removing the TLS. The strengthening of the vectorisation in the IntegrateChemistry subroutine was realised through three aspects: 1) giving the directives for the loops to instruct the compiler to vectorise the codes, including declaring no dependencies and aligning the data for efficient data accesses; 2) converting the scalar structure into vectorisation structure codes, as shown in step 3 in Figure 3, but using more complex parameter arrays to build the loop structure; and 3) in the original code segments in the IntegrateChemistry subroutine, the exponential operation sometimes was used without base-e, and these code segments had been updated to the base-e exponential operation, which can be vectorised by the AVX-512 instruction set on the KNL platform. The second part in our work was removing the TLS for the OpenMP threads. As described in Sec. 3.2, the TLS was designed to keep the data synchronisation among the threads for the common variables in Fortran. At present, the same work on the TLS is done by the compiler automatically in the way of adding codes to copy the common variables for each thread of OpenMP. The codes added by the compiler impacted the performance greatly, and it is necessary to remove the TLS. Therefore, a type structure named **cbmztype** was constructed to store the common variables. Using a PRIVATE list in the OpenMP

directive allows each thread to own a private copy of the object instance of **cbmztype**, i.e. the **cbmzobj** variable. Since **cbmzobj** is located on a thread local stack, the references to its member variables require only simple relative addressing on the stack, with simple yet efficient instructions. Meanwhile, since the common variables in the original code were no longer global and are now visible within the user subroutines, a formal parameter (argument) of **cbmzobj** was added to the subroutines using the variables.

The additional overhead of passing the address of **cbmzobj** to the subroutine is quite small. Therefore, the cost of referencing common variables in the TLS is greatly reduced with the derived type object.

Other optimisation techniques, including removing local variables to improve the memory access, were also used in CBM-Z to improve the efficiency of using caches. After all the optimisations, the CBM-Z got a 2.82× (from 3031.9 s to 1075.25 s) speedup on the CPU platform and a 3.27× (from 3031.9 s to 927.32 s) speedup on the KNL platform, respectively, as shown in Table 2.

Compared with the OpenMP performance of other modules, that of the CBM-Z module was not good enough, taking up most of the time in Opt-V GNAQMS, as shown in Figure 2, owing to the high cost of copying the rest of the common variables. More optimisations will be involved in the future to improve the OpenMP performance for CBM-Z.

### 3.2.4 Diffusion and wet deposition section

Strengthening of the vectorisation and updating of the global communication were used in the optimisation of the diffusion module.

According to the performance on the single node, the diffusion module could get a 1.99× speedup (from 241.97 s to 121.19 s) on the CPU platform and a 3.31× speedup (from 241.97 s to 73.05 s) on the KNL platform. The optimisation of the wet deposition module is relatively simple but more effective. The main optimisation of the module involves adding an OpenMP pragma to enable the multithreading for the wet deposition module. During this process, the position of allocating the private variables should be carefully chosen. The scalability of the threads in the wet deposition was really good, which allowed OpenMP to get better

performance on the KNL platform than on the CPU platform. Finally, the optimised wet deposition module got a 5.92× speedup (from 498.01 s to 84.13 s) on the KNL platform, which was much higher than the 3.19× speedup (from 498.01 s to 156.13 s) on the CPU platform.

### 4 Performance evaluation

A 48-h global atmospheric chemistry simulation was designed as the test case to test Opt-V GNAQPMS. In the test case,

GNAQPMS had full physical and chemical processes in one domain without nesting grids, which made it easier to diagnose the elapsed time. The horizontal resolution of the model was 1°×1°, which indicates that the modelling domain contained 360×180 grids. Moreover, the number of vertical layers was 20, whereas the time step for integration was 600 s in the test case. The test case was designed to test the performance of GNAQPMS on a single node of the CPU and KNL platforms and on multi-nodes of different platform clusters. This test case was an actual scientific workload and had a medium scale of calculation amount; therefore,

it allowed us to carry out much debugging and testing within a short time.

Three aspects were considered to test the performance of Opt-V GNAQPMS by comparing it with that of Base-V GNAQPMS: 1) validation of the modelling results, 2) speedup and 3) scalability, as discussed in the following section. This test case only focused on the calculation loop part except for the output part.

### 4.1 Platform setup

Intel Corporation provides a High Performance Computing environment. The Cthor Lab. of Intel Corporation was adopted for the single-node tests owing to its relatively steady environment, and Intel's Endeavor cluster was used for the cluster tests. There were

two platforms, including the CPU and KNL nodes. The CPU node had a 2.3-GHz 18-core Intel Xeon E5-2697 V4 processor, and each board contained two sockets, and its operating system was CentOS release 6.7, which is similar to the Red Hat Enterprise Linux system. The KNL node had a 1.40-GHz 68-core Intel Xeon Phi 7250 processor, and its operating system was Red Hat Enterprise Linux 7.2. The network was using the latest Intel Omni-Path Architecture (OPA). Both Base-V and Opt-V GNAQPMS

were compiled with the Intel FORTRAN Compiler 2017 Update 1, and Opt-V GNAQPMS was compiled for the CPU and KNL platforms with its own compile flags, shown in Table 3. For Opt-V GNAQPMS, the "-xCore-AVX2" and "-xMIC-AVX512" compile flags were not used for the advection module because they might cause calculation accuracy difference because of the numerical sensitivity of the advection algorithm.

## 4.2 Validation of the model results

The spatial distribution of atmospheric chemistry was used for the validation, which was plotted from the binary files outputted by GNAQPMS, as shown in Figure 5. Four species, namely, black carbon (BC), carbon monoxide (CO), ground-level ozone ($O_3$) and nitrogen dioxide ($NO_2$), were chosen to verify the model results by examining their value changes after the optimisation. According to the different reaction properties, these four species participated in different chemistry reactions. BC is a component of fine particulate matter (PM2.5), consisting of pure carbon in several linked forms, and is emitted in anthropogenic and naturally

occurring soot. In GNAQPMS, BC hardly gets involved in chemical reactions and can stay in the atmosphere for several days or even weeks. CO is spatially variable and short-lived, playing a role in the formation of $O_3$, and its spatial distribution is predominated by the emissions. $NO_2$ is one of the $O_3$ precursors, participating in the photochemical reaction with the ozone ($O_3$). Thus, CO, $NO_2$ and $O_3$ will be calculated in the gas-phase module CBM-Z of GNAQPMS. Because of this type of species diversity, the model modules can be fully covered and tested to ensure that the model results have no change during the step-by-step

optimisation. By comparing the model output results and plotting the spatial distribution images with the relative error (RE) shown in Figure 5, we can see, in the third column, that the RE was small (<1%) enough. The optimisation does not introduce an "erroneous" concentration for any atmospheric specie, and, therefore, it is reliable. However, the error could not be completely diminished because of the numerical sensitivity of the advection algorithm.

## 4.3 Speedup performance

The runtime breakdown of Base-V and Opt-V GNAQPMS on the single-node CPU platform is shown in Figure 2. In both Base-V and Opt-V GNAQPMS, the CBM-Z module played the most significant role for performance, and the absolute performance improvement for CBM-Z was remarkable after the optimisation. The better vector processing performance helped KNL to get a better speedup (3.27×) than the CPU (2.82×) in CBM-Z. However, compared with the acceleration of other modules (e.g. emission and wet deposition), that of CBM-Z was limited, which was mainly caused by the parallelisation overhead of OpenMP for the

CBM-Z module when establishing the OpenMP threads. Different combinations of OpenMP and MPI processes were tested on a single node, and the results are shown in Table 4. The speedup of the best combination for Opt-V GNAQPMS reached 3.51× on KNL and 2.77× on the CPU, compared to that of Base-V GNAQPMS, and the KNL platform had an advantageous speedup of 1.26× over the CPU platform. At the same time, without the global communication optimisation, the speedup of these combinations was 3.03× on KNL and 2.70× on the CPU. In addition, these results indicated that KNL was affected more than the CPU since the

KNL cores have a lower frequency for the I/O. The power consumption was measured with a script tool based on the Intelligent Platform Management Interface (IPMI). IPMI is a low-level interface specification that allows remote management at the hardware level without dependencies on the operating system. IPMI communicates with the server's baseboard management controller (BMC), which is a reliable agent in the system for managing and gathering system health, including power consumption data. The

average power was 440 W and 324 W for the CPU and KNL platforms, respectively. Therefore, the average power of KNL was 26% lower and the average energy consumption was 37.5% lower than those of the CPU platform. The faster speed and lower energy consumption enabled KNL to outperform the CPU on a single node.

### 4.4 Scalability on a cluster

The cluster performance of the atmospheric model was measured by strong scalability. Strong scalability indicates how many computing resources can be used when the workload is fixed, which can be measured by the speedup with increasing node number. Better scalability means the model can use more computing resources to deal with a task and complete the task at a shorter time. The scalability was measured by recording the speedup of the core calculation portion of the model on clusters.

As shown in Figure 6, Base-V GNAQPMS could maximally use eight two-socket CPU nodes for the test case. After the
optimisation, the parallel scalability of GNAQPMS was greatly improved on both the CPU and the KNL cluster, scaling to 40 CPU nodes and 30 KNL nodes, with a parallel efficiency of 70.4% and 42.2%, respectively. Opt-V GNAQPMS could use more than 40 two-socket CPU nodes for the same test case. The test of scalability on the CPU cluster was constrained by the limited computing resources, and the scalability could be expected to extend to more than 40 nodes. However, the scalability curve of the KNL cluster was lower than that of the CPU cluster, as shown in Figure 6; moreover, Opt-V GNAQPMS could only use 30 KNL
nodes at most. On a single node, Opt-V GNAQPMS had a higher performance on the KNL platform than on the CPU platform when the number of nodes reached 12; however, its speed on the KNL platform was lower than that on the CPU platform. This was mainly caused by too many MPI processors and the not-good-enough performance of the GNAQPMS OpenMP code segments on KNL, which has 68 cores, almost four times more than those of the CPU. According to the above test, further optimisation of OpenMP is needed to improve the cluster performance of Opt-V GNAQPMS on the KNL cluster. In addition, Figure 7 shows that
the simulation speed, or model second per real second, improved with increasing number of nodes. When reaching 40 CPU nodes, the CPU cluster could do the simulation of 8 model years per day, and with 30 KNL nodes, it could do the simulation of 3.7 model years per day excluding the I/O part. The optimisation work in this study has made the computation performance of GNAQPMS very close to our anticipated goal of 5-10 model years under the coarse spatial resolution.

### 5 Conclusions

In this study, the global chemistry transport model GNAQPMS was optimised to run on the Intel second-generation MIC architecture KNL processor and accelerate its modules. The main optimisation methods and tips were used including 1) updating the pure MPI parallel mode to the hybrid parallel mode with MPI and OpenMP; 2) strengthening the vectorisation by constructing loops and using compiler directives in GNAQPMS to make full use of the 512-bit-wide VPU on the KNL platform; 3) reducing unnecessary memory access to improve the utilisation efficiency of caches; 4) removing the TLS for common variables with each
OpenMP thread to improve the OpenMP efficiency; and 5) changing the way global communication works from writing/reading interface-files to using MPI functions.

The tests of Opt-V GNAQPMS were conducted on the latest Xeon E5-2697 V4 and KNL 7250 clusters. Both single-node and multi-node cluster performances were tested. For the single node, Opt-V GNAQPMS achieved a speedup of 2.77× on the CPU platform and a speedup of 3.51× on the KNL platform compared with the Base-V model on the CPU platform. The power and
energy consumption of KNL was 26% lower than that of the CPU. With the combined performance and energy improvement, the KNL platform was 37.5% more efficient in terms of power consumption compared with the CPU platform. Compared with the CPU platform, the KNL platform had obvious advantages such as fast speed and lower energy consumption. The cluster test results

showed that the scalability of GNAQPMS on the CPU platform was largely increased from 8 nodes to up to 40 nodes, whereas that on the KNL platform was not as good as that on the CPU platform owing to the bottleneck of the MPI global communication and fragmental OpenMP parallel regions. In summary, the computation speed (excluding the model I/O) was improved from about 0.5 model years per day using 8 CPU nodes to about 3.7 model years per day using 30 KNL nodes and about 8 model years per day using 40 CPU nodes, respectively. Therefore, without regard to the model I/O, the optimisation work in this study resulted in the computation performance of the GNAQPMS being very close to our anticipated goal. In the next step, further work will be focused on merging the OpenMP parallel regions. Moreover, the I/O optimisation was not considered in this study and it should be taken into account in the future.

The general suggestions we could give for the optimisation of other models on KNL are as follows: 1) the coder should focus on the vectorisation of codes, and 2) as the performance of OpenMP is very important for KNL, the coder should design more efficient parallel regions for OpenMP.

**Code availability**

The source codes of Base-V and Opt-V GNAQPMS are available online via ZENODO (https://doi.org/10.5281/zenodo.290203) at doi:10.5281/zenodo.290203.

**Acknowledgements:**

The National Natural Science Foundation of China (41305121 and 41405119), the National Major Research High Performance Computing Program of China (2016YFB0200800) and the Environmental Public Welfare Research Project of China (201509014) funded this work. The authors would like to thank Intel Corporation's Software Support Group (SSG) for providing the High Performance Computing (HPC) environment and technical support for the case testing of GNAQMS. Special thanks should go to Victor Lee for providing the professional English writing instruction.

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

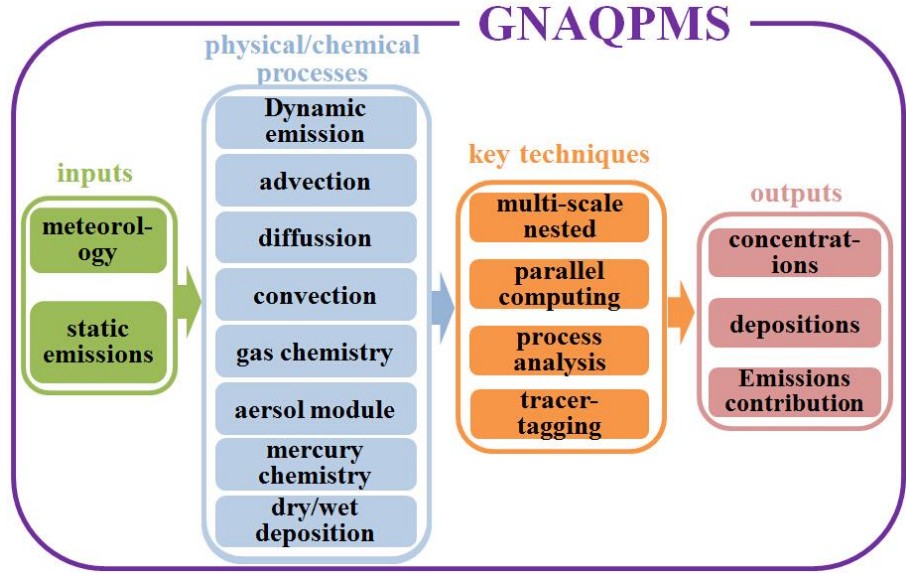

**Figure 1. Framework of the Global Nested Air Quality Prediction Modeling System (GNAQPMS)**

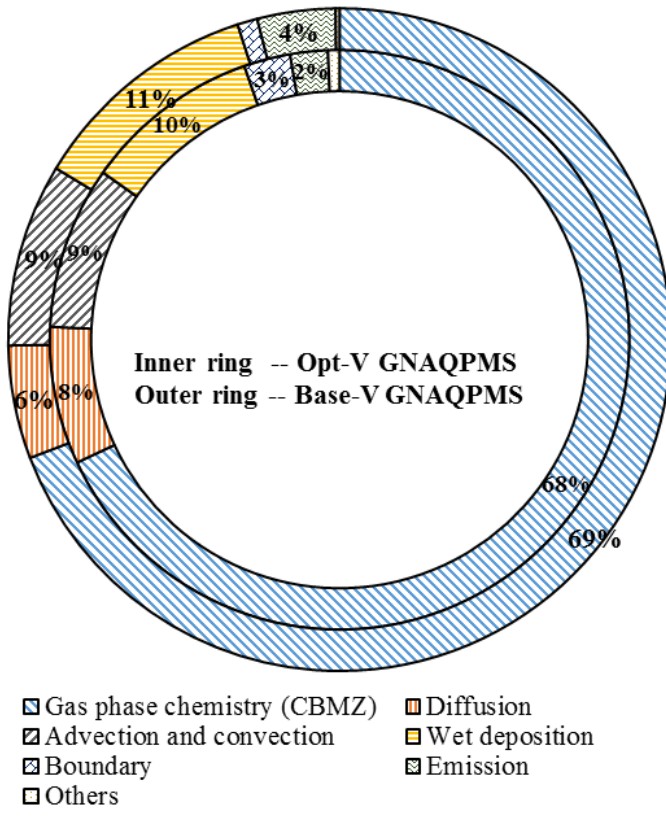

Inner ring  -- Opt-V GNAQPMS
Outer ring  -- Base-V GNAQPMS

☒ Gas phase chemistry (CBMZ)   ▥ Diffusion
▨ Advection and convection     ▨ Wet deposition
▨ Boundary                     ▨ Emission
▢ Others

Figure 2. Overhead proportions of Base-V GNAQPMS (outer ring) and Opt-V GNAQPMS (inner ring). The top five most time-consuming modules were CBM-Z, diffusion, advection, wet deposition and boundary exchange.

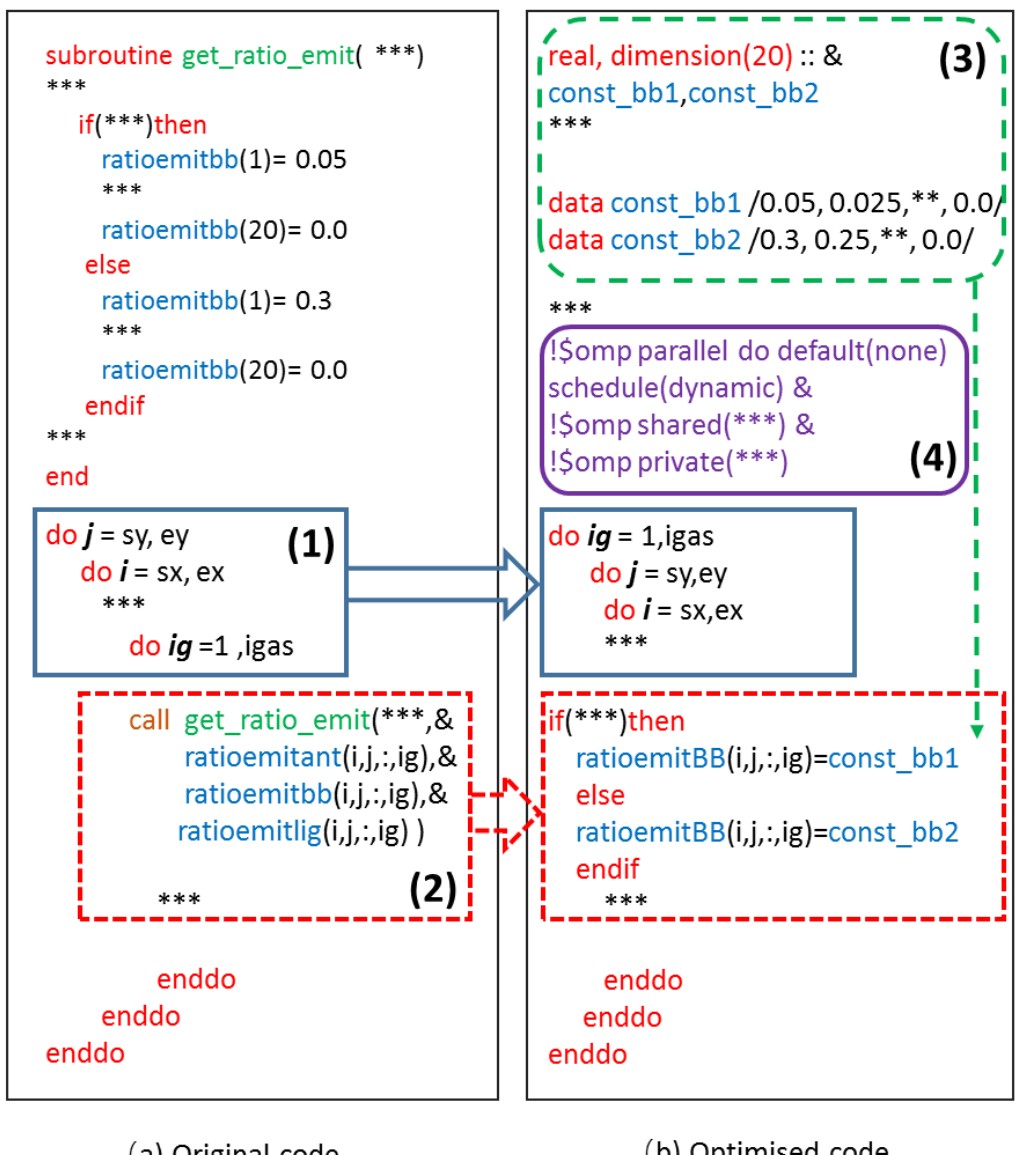

**Figure 3. Sample codes containing some typical optimisation methods. Part (a) is the original code and part (b) is the optimised code. Step (1) changes the order of the i, j, ig loops, and step (2) makes the subroutine inline. Step (3) uses the parameter to convert the scalar codes to vector codes, and step (4) adds the OpenMP pragmas.**

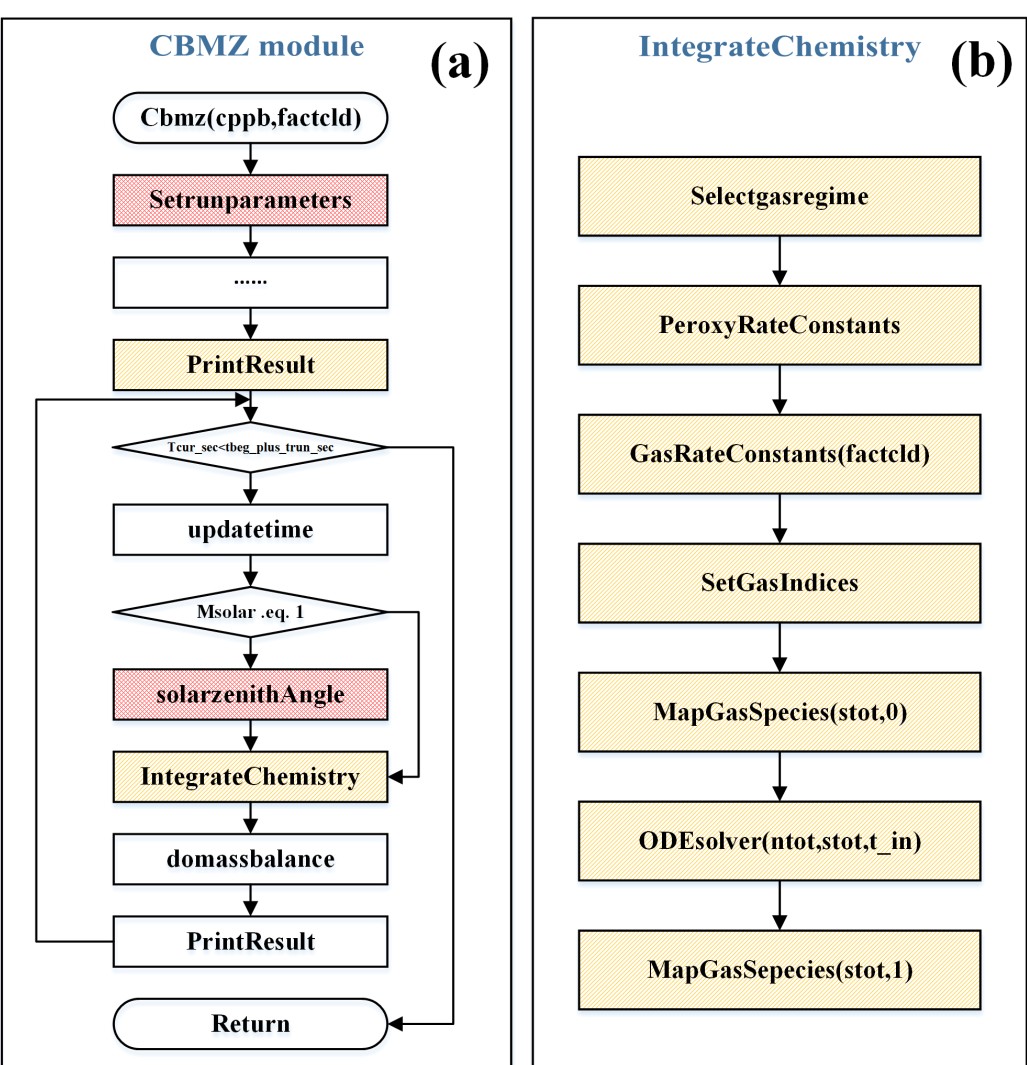

**Figure 4. Flowcharts of the CBM-Z module (a) and the IntegrateChemistry subroutine (b). The red subroutines were removed and made into inline functions, and the orange parts were modified for vectorisation.**

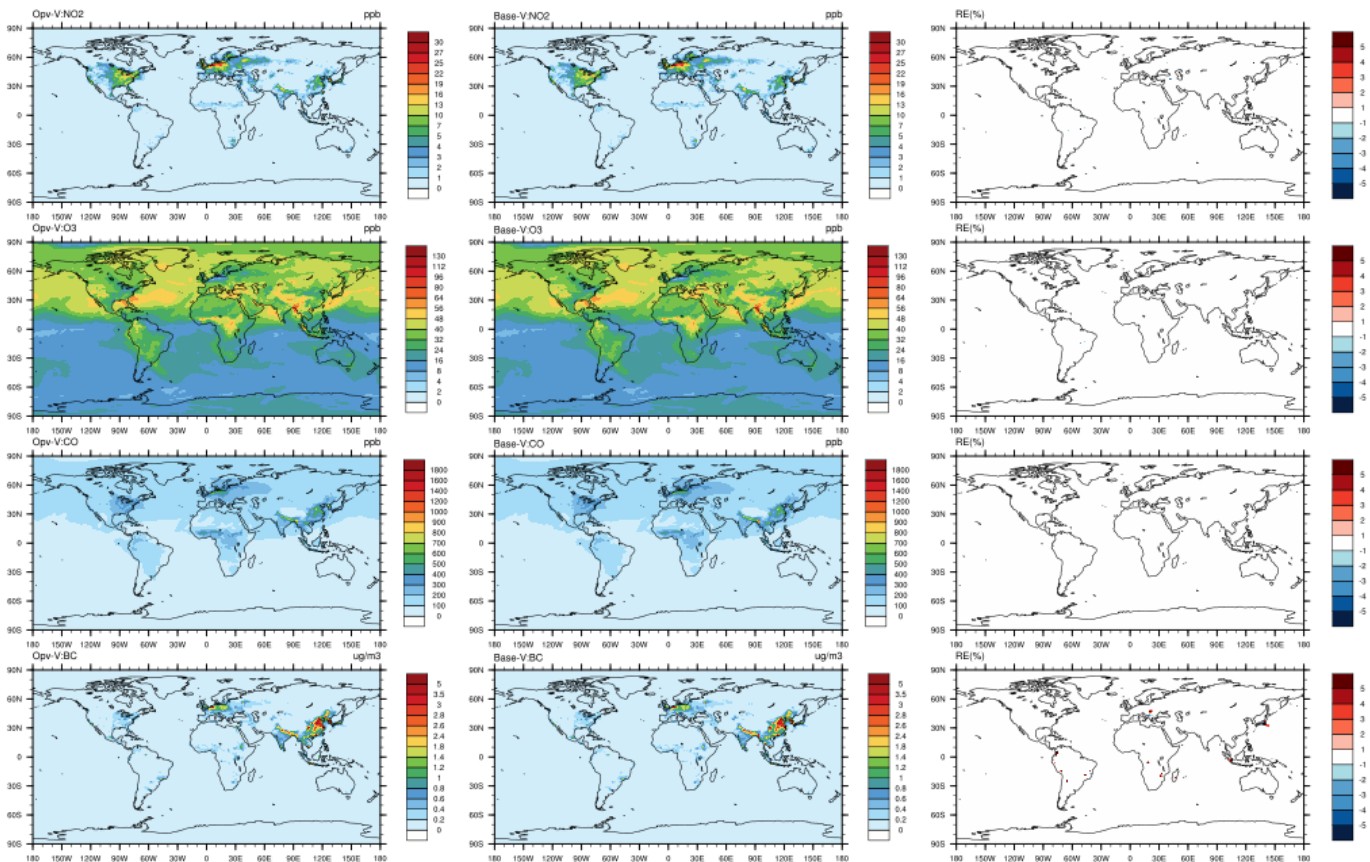

**Figure 5. Spatial distribution of BC, CO, O₃ and NO₂ from Opt-V and Base-V GNAQPMS.**

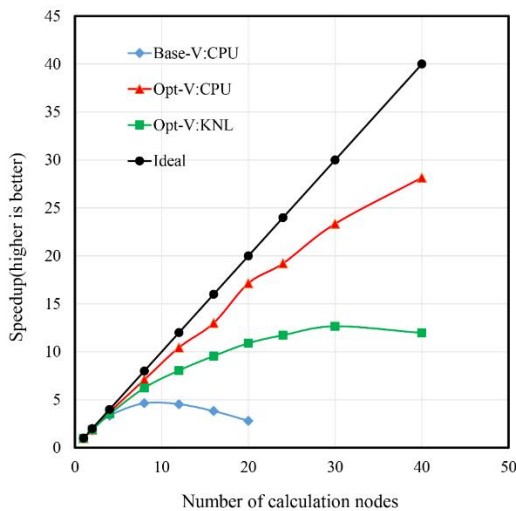

**Figure 6. Scalability of Base-V and Opt-V GNAQPMS on the CPU and KNL clusters. Base-V GNAQPMS on the CPU cluster had a bad scalability and its performance was nearly saturated on 8 nodes, whereas Opt-V GNAQPMS could reach at least 40 nodes on the CPU and at most 30 nodes on KNL.**

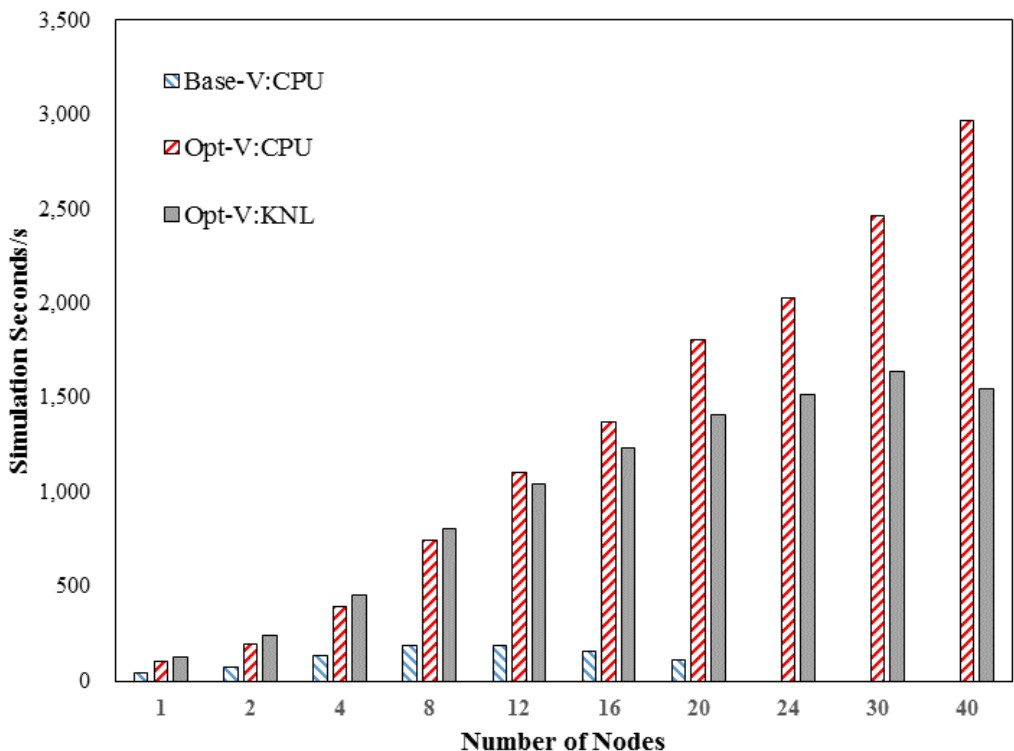

**Figure 7. Simulation ability improved with increasing number of nodes.**

**Table 1. General indicators detected by the VTune HPC performance detection tool for the two versions of GNAQPMS.**

| Version | SP GFLOPS | CPU Utilisation | Memory Bound | FPU Utilisation |
|---------|-----------|-----------------|--------------|-----------------|
| **Base-V** | 93.741 | 82.5% | 9.2% | 2.9% |
| **Opt-V** | 279.479 | 74.1% | 12.7% | 9.6% |

5 **Table 2. Optimisation measures for the main modules and the speedup after the optimisation on the CPU and KNL platforms. The well-parallelised modules (e.g. emission module) could get a high speedup of 12.83× on KNL, and a 3.27× speedup on KNL and 2.82× speedup on the CPU were achieved for the CBMZ gas-phase chemistry module, which was the most time-consuming part (68%) for the baseline version.**

| Items | OpenMP Optimisation | Strengthen the Vectorisation | Remove TLS | Change Global Communication | Remove Redundant Calculation and Memory Access | Speedup on the CPU | Speedup on KNL |
|-------|---------------------|------------------------------|------------|------------------------------|------------------------------------------------|--------------------|----------------|
| Initialisation | | | | Yes | | 1.27 | 1.26 |
| Emission | Yes | Yes | | Yes | Yes | 5.03 | 12.83 |
| Advection and convection | Yes | Yes | | Yes | Yes | 2.72 | 3.18 |
| Diffusion | | Yes | | Yes | | 1.99 | 3.31 |
| Gas-phase chemistry (CBM-Z) | Yes | Yes | Yes | Yes | Yes | 2.82 | 3.27 |
| Wet deposition | Yes | | | Yes | | 3.19 | 5.92 |

**Table 3. Compile flags of Base-V GNAQPMS and Opt-V GNAQPMS on the CPU and KNL platforms**

| VERSION OF GNAQPMS | INTEL COMPILER FLAGS |
|---|---|
| **BASE-V** | -O3 –init=arrays –init=zero –fpp –w – traceback –ftz –fno-alias – fno-fnalias -g |
| **OPT-V (CPU PLATFORM)** | –O3 –ip –init=arrays -xCore-AVX2 –fp-model fast=1 –O3 –ip – init=arrays –init=zero –qopenmp –ftz –fno-fnalias –fno-alias –g – w – traceback |
| **OPT-V (KNL PLATFORM)** | –O3 –ip –init=arrays –MIC-AVX512 –fp-model fast=2 -align array64byte –qopenmp –ftz –fno-fnalias –fno-alias –g –w – traceback |

**Table 4. Speedup and wall time of different combinations of OpenMP threads and MPI processes.**

| CPU(E5-2697 V4 with 36 physical cores and 2 hyper-threads) | | | | |
|---|---|---|---|---|
| | OMP | MPI | Wall Time | Speedup |
| Baseline (no hyper-thread) | 0 | 36 | 4381.2 | 1 |
| Opt-V | 1 | 72 | 1769 | 2.48 |
| | 2 | 36 | 1625.72 | 2.70 |
| | 4 | 18 | 1614.9 | 2.71 |
| | 6 | 12 | 1580.1 | 2.77 |
| | 12 | 6 | 1612.3 | 2.72 |
| | 18 | 4 | 1790.2 | 2.45 |
| | 36 | 2 | 2243.4 | 1.95 |
| Opt-V (no global communication) | 6 | 12 | 1623.6 | 2.70 |
| KNL (KNL 7250 with 68 physical cores and 4 threads) | | | | |
| Opt-V | 2 | 136 | 1499.2 | 2.92 |
| | 4 | 68 | 1402.9 | 3.12 |
| | 2 | 68 | 1512.8 | 2.90 |
| | 4 | 34 | 1248.3 | 3.51 |
| | 8 | 34 | 1373.6 | 3.19 |
| | 16 | 17 | 1473.2 | 2.97 |
| Opt-V (no global communication) | 4 | 34 | 1444.6 | 3.03 |