# Peer review of "GNAQPMS v1.1: Accelerating the Global Nested Air Quality Prediction Modeling System (GNAQPMS) on Intel Xeon Phi Processors"

_Geoscientific Model Development, 2016_

## Short Comment (SC1) · 24 Feb 2017

Dear authors,

in my role as Executive editor of GMD, I would like to bring to your attention our Editorial version 1.1:

http://www.geosci-model-dev.net/8/3487/2015/gmd-8-3487-2015.html

This highlights some requirements of papers published in GMD, which is also available on the GMD website in the 'Manuscript Types' section:

http://www.geoscientific-model-development.net/submission/manuscript_types.html

In particular, please note that for your paper, the following requirement has not been met in the Discussions paper:

- "The main paper must give the model name and version number (or other unique identifier) in the title."

Please add a version number for GNAQPMS in the title upon your revised submission to GMD.

Yours,

Astrid Kerkweg

---

## Short Comment (SC2) · 4 Mar 2017

Dear Astrid Kerkweg,

Thanks a million for your precious time and kind reminding. This work is based on the GNAQPMSv1.0 model (Chen et al, 2015). According to the discussion of all authors, the version number of the GNAQPMS model would be "v1.1", because this work focused on the computing performance and the model framework hasn't been changed. In addition, the title would be "GNAQPMS v1.1: Accelerating the Global Nested Air Quality Prediction Modeling System (GNAQPMS) model on Intel Xeon Phi processors" in revision manuscript.

Sincerely Yours.

Wang Hui

Reference

Chen H S, Wang Z F, Li J, et al. GNAQPMS-Hg v1. 0, a global nested atmospheric mercury transport model: model description, evaluation and application to trans-boundary transport of Chinese anthropogenic emissions[J]. Geoscientific Model Development, 2015, 8(9): 2857-2876.

---

## Referee Comment (RC1) · Anonymous Referee #1 · 20 Mar 2017

General Comments

The paper describes the work done by the authors on optimizing the GNAQPMS model for Intel Knights Landing. The paper is well organized. I agree with the authors that there are still options to further optimize the code but the work performed so far is a good first step and worth publishing. There are a few changes that need to be addressed. Also the language could use some general improvement. I will first give some specific comments that I believe need to be addressed before publication. After that I list a few (optional) suggestions for future optimizations.

Specific Comments

(a) section 1, page 3: according to the introduction Section 3.1 is supposed to discuss where the bottlenecks of the code are. I assume that this refers to the run-time measurements shown in Figure 2? Under the term "bottleneck" I would have expected a discussion whether the code is bound by memory bandwidth by showing measurements of memory bandwidth and flops/s and comparing them with the peak performance obtained for the STREAM and LINPACK benchmarks. I understand that hardware counters are not very accurate on Intel architectures but you could still count them with an emulator or by hand. The paper does not show in its current state what the bottlenecks of the different parts of the code are.

(b) section 4.2, page 9: comparing the patterns and values without giving any specific relative difference between base and optimized version of the code is not enough to claim that the results are identical. The authors need to show some precise numbers like the total mass of the air and the different chemical species in both versions and the difference between optimized and base version.

(c) section 4.3, page 9: how did you measure the power consumption? Do you trust VTune to give you these measurements or did you measure the power consumption yourself?

(d) How many OpenMP threads were used per MPI process? Did you try different configurations (like 2, 4, 8, 16, 32, 64 threads per MPI process)?

(e) The term "manual vectorization" is used many times throughout the paper. This is very misleading. Manual vectorization would be in my opinion if the code was rewritten with avx512 vector intrinsics in C! The vectorization used in this paper still relies on the compiler.

Suggestions

(f) As mentioned before the paper does not investigate what the real bottleneck of the code is and how far the code is from optimal performance. I highly recommend to

create a theoretical and measured roofline model. This would allow to answer how much potential for further optimization should still be possible.

(g) I would love to know how many of all floating point operations are vectorized. I understand that counting floating point operations on Intel architectures through hardware counters is not very accurate. Maybe you could give some estimates from what the vectorization report shows you?
* * *

---

## Referee Comment (RC2) · Anonymous Referee #2 · 25 Mar 2017

The manuscript presents results of efforts to improve the performance and scalability of a global atmospheric chemistry model, the QNAQPMS, using the Knights Landing version of the Intel Xeon Phi. Gains are relative to the original unoptimized code on both KNL and on Broadwell, Intel's latest-generation multi-core Xeon processor. The subject of the work, increasing simulation capability of QNAQPMS using next-generation processors to improve computational performance and scaling, represents a noteworthy contribution to modeling science within the scope of Geoscientific Model Development. The contribution comes in the form of new concepts (next generation processors) and methods (measuring performance, identification of bottlenecks, and optimization through restructuring of application code). The approach and methods

appear to be valid and reasonable, and the references to related work are appropriate and sufficient.

This reviewer did not evaluate the manuscript with respect to scientific reproducibility. The paper appears to present sufficient detail on the workload, configuration of the application, and the computational platforms tested that another scientist with access to the code, datasets and computer resources would be able to reproduce the performance and scaling results presented. However, the verification of model output from the optimized version appears to be based on merely eyeball comparisons of color contour plots. The lack of a suitable objective verification technique would make it impossible to be sure the results were reproduced.

Regarding the presentation of the results, the clarity and the level of detail are very poor. There are significant gaps in the explanation of, for example, the analysis of hotspots and bottlenecks in the code, and their relative impacts on performance.

The use of VTune is mentioned, but how do the authors determine whether a particular hotspot is performing inadequately, and if so, what is the specific bottleneck? Insufficient use of vectors? Hardware threads? Memory bandwidth or latency? Load imbalance?

The use of OpenMP threading is also insufficiently motivated or explained.

This reviewer also takes issue with including "changing global communication from interface-files writing/reading to MPI functions" among the list of five optimizations discussed, since this change appears to address a fundamental deficiency in the parallel design of the original application, not one specific to optimizing for the Intel Xeon Phi processor. This reviewer suggests removing discussion of this optimization from the paper and using the space to more fully explain the other four optimizations. If, however, the authors wish to include discussion and results of the global communication optimization in this paper, its effect on performance and scaling should be clearly separated from the other optimization results. As currently presented, the reader cannot

deterimine what effect this has relative to the effects of the KNL-specific optimizations.

Because of these problems, which go beyond language and grammatical issues, this reviewer recommends the paper be rejected and reconsidered only after significant revision.

Specific comments:

Pages 1 and 2, Introduction. Discussion of impacts is incomplete. What levels of performance does QNAQPMS currently provide for simulation science using the model? What scientific problems are currently possible? What scientific problems are beyond reach with current QNAQPMS performance and scaling. How much more performance will be is required to enable these larger problems. Be specific.

Page 3, Section 2.1, Model Description of GNAQPMS. (Suggestion) What is the difference between GNAQPMS and NAQMPS? Basing the description of GNAQPMS on the NAQPMS presupposes knowledge about the NAQPMS. There is already a reference to NAQPMS but also add a few sentences of background on the NAQPMS. Explain why the authors focus GNAQPMS and not at the NAQPMS.

Page 4, line 3: "Since the memory processor is not dominant". Unclear, explain further: what do the authors mean specifically by "memory pressure". Memory working set size? Memory bandwidth requirement of this application. How have they determined it is not dominant?

Page 4, line 9: Spell out first use of TLS to represent thread local storage. It is spelled out in the abstract, but needs to be spelled out again in main body of paper.

Page 5, paragraph beginning line 6 and all of Section 3.2.1. As noted in the general comments above, the use of files for global communication instead of MPI reductions and gathers is problematic for any modern parallel architecture, not just the KNL. Strongly recommend removing discussion of this optimization, including the entire Section 3.2.1, from the paper. It is relevant to optimizing for KNL.

Page 6, line 11. "Cyclic order"? Do the authors mean loop nesting order?

Page 6, line 12. "We cancelled the calling of the subroutine ... and made it an internal function in main program." This technique is referred to as inlining. Did the authors look at inlining reports from the Intel compiler to see if this could have been accomplished automatically or with the help of directives and compiler options, without manually restructuring the code? Also, the authors mean to say the "calling subroutine", not the "main program".

Page 6, line 14. "... using parameters to convert scalar structure to vector structure." Unclear what "coverting scalar to vector" means, nor does it appear from step 3 Figure 3 that this is what is going on.

Page 6, line 15. "... we added the directives, clauses, declarations and syntax comment of OpenMP outside the outermost loop as shown in box 4." This sentence appears to be the only discussion in the paper of how and why OpenMP threading was added to the code. Given that this is one of the five optimizations being presented, this is insufficient. It's not clear why the authors felt it was important to add OpenMP threading nor whether, from the results, it provided a benefit.

Page 7, line 15. "2) updating some code segments to the the serial codes to construct vectorization." What does this mean?

Page 7, lines 17-25. A reader might make a number of educated guesses about what the authors are saying was the bottleneck and how it was fixed, but it is not clear at all from the text. What codes were "added by the compiler?" Was the issue a copy-in/copy-out problem for thread-local variables that were listed as private/firstprivate in the OpenMP directives? If so, how did adding the CBZOBJ argument fix this? By passing-by-reference? If so, how were data races avoided? Were they avoided because the CBMZOBJ objects themselves were THREADPRIVATE? There's a lot of important detail missing from this discussion.
Page 7, line 32. "...which spent 10 percentages and 8 percentages." One would prefer to see actual timings instead of percentage of runtime in discussion of performance improvement.

Page 7, line 36. "1.78 speedup on the CPU platform and 2.39 speedup on the KNL." How much of these speedups were from the manual vectorization and how much from the optimization of global communication? (Again, this reviewer suggests not considering global communication optimization at all, but if it is discussed, show effects separately from other optimizations).

Page 8, lines 7-10. "The horizontal resolution of the model is $1° \times 1°$, which indicates that the modelling domain contains $360 \times 180$ grids. And the number of vertical layers is 20, while the time step for integration is 600 seconds in the test case. The test case was designed to test the performance of GNAQPMS on single node of CPU and KNL platform, and multi-nodes on different platform clusters." This workload is very small and probably not suitable for KNL clusters, which require a high degree of parallelism to be efficient. (The authors make note of this on page 9, lines 32-33). The paper describe a representative workload for scientific simulations using the GNAQPMS and provide some discussion of how the performance results presented are relevant to an actual scientific workload.

Page 8, line 15. "The Intel Corporation provides the High Performance Computing environment for the test." Was this Intel's Endeavor cluster? Perhaps mention this as well.

Page 8, line 20. Opt-V GNAQPMS has been compiled on CPU and KNL platform, respectively." Not sure what this means. Are the authors actually compiling the codes on the respective platforms? If so, why? Is there some difference expected between a native-compiled and a cross-compiled executable?

Page 8, line 21. "... the -xCore-AVX2 and –xMIC-AVX512 compile flags were not used for the advection module..." This is troubling and bears further discussion. What did the

authors see that caused them to avoid these compiler flags for the advection module? Why is advection susceptible to differences but not other parts of the code such as the ODE solver?

Page 9, line 2. "...were confirmed to be identical." This is not a persuasive verification method. An eyeball comparison of plots would not be considered sufficient to confirm that results are identical. Provide difference plots and RMS difference statistics.

Page 9, line 20. "...when the computing scale is fixed." Right word? Instead maybe workload? problem size?

Page 9, line 24. "After optimization, the parallel scalability of GNAQPMS is greatly improved..." This seems counterintuitive, since vectorization and other node-performance optimizations that improve performance on each node from should make the code less well, assuming that interprocessor communication overheads are the same. It's important to distinguish here (1) what parts of the code are being timed for the scaling measurements. (2) Which optimizations are contributing to the improved scalability and which may be improving performance but working against strong scaling.

Page 9, line 33. "... OpenMP code segments on KNL..." What is the effect of varying the number of OpenMP threads with respect to MPI tasks? That is, using the same overall number of hardware threads across the job? What is the effect of running pure MPI? Pure OpenMP?

Page 10, Conclusion section. Future work and areas for improvement are discussed. Howeve,r the conclusion should also include discussion of what levels of performance and scaling are needed for simulation science using GNAQPMS. The conclusion should assess whether and how well the presented work helps the GNAQPMS users achieve these goals.

Figure 7. Shows speedup but not performance. Suggest adding a figure that shows performance as a function of the number of nodes. Plot as simulation seconds per

second of wall clock time.

---

## Author Comment (AC1) · 18 May 2017

**Dear reviewer,**

Thanks a million for your precious time and comments. We will reply the reviewer's comment point-to-point in the following part.

**"The manuscript presents results of efforts to improve the performance and scalability of a global atmospheric chemistry model, the QNAQPMS, using the Knights Landing version of the Intel Xeon Phi. Gains are relative to the original unoptimized code on both KNL and on Broadwell, Intel's latest-generation multi-core Xeon processor. The subject of the work, increasing simulation capability of QNAQPMS using nextgeneration processors to improve computational performance and scaling, represents a noteworthy contribution to modeling science within the scope of Geoscientific Model Development. The contribution comes in the form of new concepts (next generation processors) and methods (measuring performance, identification of bottlenecks, and optimization through restructuring of application code). The approach and methods appear to be valid and reasonable, and the references to related work are appropriate and sufficient."**

**Reply:** Thanks for the praise from the reviewer. We believe this work could be a good example for the model developer who wants to transplant their model to KNL platform. However, we still need to emphasize that the name of our model is GNAQPMS, not QNAQPMS called by reviewer. To ensure that this name is not coming from our manuscript, we checked our discussion paper and we did not find such name.

**"This reviewer did not evaluate the manuscript with respect to scientific reproducibility. However, the verification of model output from the optimized version appears to be based on merely eyeball comparisons of color contour plots. The lack of a suitable objective verification technique would make it impossible to be sure the results were reproduced."**

**Reply:** Thanks for the comments. We designed two mechanisms to validate the results. On one hand, we added an extra module to test our results. The function of this module is to output the concentration of specific chemical species after each chemical or physical processes finished. Each process writes its own data into its own files respectively at the same time, and each chemical and physical module would only run one time to insulate the effect of other modules. Then, an additional small program will read the output files from the two version of GNAQPMS to check and report the relative and absolute errors. This method is likes that sampling the

results during the running period. In this way, we can find the main error between each sections. On the other hand, we could plot the spatial distribution of the two model results after a long time integration, and the Figure 5 and Figure 6 with Relative Error (RE) and Relative Mean Square Difference (RMSD). And RE of almost all girds is lower 1%, which is acceptable for us. However, we can't diminish the error currently because of the numerical sensitivity of the advection.

"**Regarding the presentation of the results, the clarity and the level of detail are very poor. There are significant gaps in the explanation of, for example, the analysis of hotspots and bottlenecks in the code, and their relative impacts on performance.**

**The use of VTune is mentioned, but how do the authors determine whether a particular hotspot is performing inadequately, and if so, what is the specific bottleneck? Insufficient use of vectors? Hardware threads? Memory bandwidth or latency? Load imbalance?"**

**Reply:** Thanks for the comments. The reviewer required to present more details about the processes of identifying the bottlenecks and optimization methods. We agreed with the reviewer and we will do the relevant modification in the revised manuscript.

Overall, three steps were used to complete the modernization of model codes: 1) Performance profiling; 2) Single node optimization; 3) Multi nodes optimization.

The bottleneck identification in this study are based on the common way like the Vtune tools as well as the application behavior analysis and characteristic extraction. The Vtune, ITAC as well as advisor tools can identify most of the problems mentioned by the reviewer. The ITAC is mainly used to analyze the MPI performance and communication problems, including the time consuming of the MPI functions and OMP threads. Figure 1 shows the graphic report of the load balance situation of GNAQPMS (using 72 MPI processes), which comes from the ITAC, and it illustrates that the load balance is not the main problem of GNAQPMS.

[Figure]

**Figure 1 Load balance situation of GANQPMS on single CPU nodes.**

The Vtune tools currently could only be used on single node, however, it contains more functions to help us to identify the bottleneck. The reviewer mentioned the bandwidth problem, and Vtune could collect the data as well as finish the visualization as showed in Figure 2.

[Figure]

**Figure 2 the bandwidth situation during the application running.**

The Vtune detects the hotspot functions. Figure 3 presents the hotspots in the Base version GNAQPMS detected by Vtune, and the red bar indicates the low CPU usage efficiency. The list of hotspots would clarify the priority of optimization work. Moreover, Figure 4 shows the hotspots in the current version of Opt-V GNAQPMS with better CPU usage efficiency. And the solution for the hotspots depends on the analysis of the codes of the hotspots. The Intel Advisor could help to do the vectorization and multi-threads work. And bandwidth issues could be detected by the Vtune by analyzing the memory-access.

[Figure]

**Figure 3 hotspot functions detected by Vtune in Base-V GNAQPMS.**

[Figure]

**Figure 4 hotspot functions detected by Vtune in Opt-V GNAQPMS**

"**The use of OpenMP threading is also insufficiently motivated or explained.**"

**Reply:** Thanks for the comments. As to OpenMP, our concern is mainly about the following two aspects. On the one hand, our goal is to accelerate the model and improve the scalability, and our primary desire is to replace the MPI processes by the relative cheap OpenMP threads as many as possible on each calculation node, which is adopted in other large-scale models, such as CESM. On the other hand, considering the features of multi-core and low frequency of KNL, the large number of pure MPI processes would lead to expensive communication cost for KNL and it could be inevitable to do the hybrid optimization. And following description is also added to the In Page2 Line 35:

"KNL contains many low frequency cores, and each core contains four hyper threads. Considering the relative expansive overhead of communication of MPI for many cores of KNL, we adopted the hybrid parallel mode by using OpenMP and MPI. Furthermore, the OpenMP threads could fully use the hyperthreads in KNL, and the cheap communication cost of OpenMP could help to improve the scalability"

**"This reviewer also takes issue with including "changing global communication from interface-files writing/reading to MPI functions" among the list of five optimizations discussed, since this change appears to address a fundamental deficiency in the parallel design of the original application, not one specific to optimizing for the Intel Xeon Phi processor. This reviewer suggests removing discussion of this optimization from the paper and using the space to more fully explain the other four optimizations. If, however, the authors wish to include discussion and results of the global communication optimization in this paper, its effect on performance and scaling should be clearly separated from the other optimization results. As currently presented, the reader cannot deterimine what effect this has relative to the effects of the KNL-specific optimizations."**

**Reply:** Thanks for the comments. We should declare that none of the optimizations adopted in this paper is specific for the KNL platform. Moreover, performance portability is one of the advantages of KNL. Once the code modernization work is done for KNL, the performance can be easily ported to new generation Intel CPU (e.g. Broadwell, Skylake) sharing hardware features with the same code.

As mentioned by the reviewer, the interface-files could be the problematic for any parallel architecture, but this problem is part of our work to deal with. Moreover, this paper describe the common measures as well as the special measures for our model, and low frequency cores in KNL lead to more I/O overhead on KNL compared with CPU, which is indicated by the effect on speedup of KNL in Table 1(from 3.51 to 3.03). Since the overhead of file I/O increases much faster than that of collective MPI functions as the number of nodes increases, this optimization could be the key to the scalability. However, the Endeavor cluster has updated its nodes configuration, and the benchmark CPU (E-5 2697 V4) nodes have been replaced. Therefore, we could not provide the scalability effects based on the previous test results.

**"Because of these problems, which go beyond language and grammatical issues, this**

**reviewer recommends the paper be rejected and reconsidered only after significant revision."**

**Reply:** Thanks for your comments. As for the language and grammatical issues, we will invite the native English speaker or some relative company to do the copy editing for English.

According to the reviewer's specific comments, the following modifications have been done.

1. **Pages 1 and 2, Introduction. Discussion of impacts is incomplete. What levels of performance does QNAQPMS currently provide for simulation science using the model? What scientific problems are currently possible? What scientific problems are beyond reach with current QNAQPMS performance and scaling. How much more performance. Be specific.**

**Reply:** Thanks for the comments. GNAQPMS is designed for global atmospheric aerosol and chemistry simulation. Its applications include temporal and spatial evolution of atmospheric composition (ozone, black carbon, sulphate, nitrate, dust, seasalt et al.), providing boundary conditions for regional models, intercontinental long-range transport, long-term climate change (aerosol-cloud-radiation interaction), and it also acts as a key component of the Earth System Model of Chinese Academy of Sciences (CAS-ESM). Currently, GNAQPMS can only run on CPU platform, and its parallel scalability and computation speed are about 8 CPU nodes and 46 hours per model year at 1°x1° resolution (excluding model I/O). This model performance is suitable for short-term or medium-term (5 years or less) simulation but not suitable for long-term simulation (30 years or more, needs more than 2 months of computation time). Besides, the model computation time will further increase when model I/O is included or higher model resolution (e.g. 0.25°x0.25°) is used. Therefore, it cannot be directly coupled into earth system model and used for long-term climate change simulation. By optimization of model codes and usage of new hardware, we aim to greatly improve the model parallel scalability and computation speed. The target computation speed in the future is about 5-10 model years per day (including model I/O) at 0.25°x0.25° resolution. This is an ambitious goal and needs a lot of hard work. We plan to improve the parallel computation speed of GNAQPMS in the first step, and improve the model I/O efficiency in the next step.

The Introduction (the fourth paragraph) of the manuscript was revised accordingly to make the

impact and purpose of this study more specific.

**2. Page 3, Section 2.1, Model Description of GNAQPMS. (Suggestion) What is the difference between GNAQPMS and NAQMPS? Basing the description of GNAQPMS on the NAQPMS presupposes knowledge about the NAQPMS. There is already a reference to NAQPMS but also add a few sentences of background on the NAQPMS. Explain why the authors focus GNAQPMS and not at the NAQPMS.**

**Reply**: Thanks for the suggestions. The GNAQPMS model is a global multi-scale chemical transport model based on the Nested Air Quality Prediction Modeling System (NAQPMS) (Wang et al., 2006), developed at the Institute of Atmospheric Physics, Chinese Academy of Sciences. NAQPMS is a 3-D regional Eulerian model which has been rigorously evaluated and widely applied to simulate the chemical evolution and transport of ozone (Li et al., 2007; Tang et al., 2010), the distribution and evolution of aerosol and acid rain over East Asia (Wang et al., 2002; Li et al., 2011; Li et al., 2012) and to provide operational air quality forecasts in mega cities such as Beijing, Shanghai and Guangzhou (Wang et al., 2010; Wu et al., 2012; Wang et al., 2009). GNAQPMS and NAQPMS use the similar model framework, physical and chemical parameterization schemes and parallel computation techniques. The optimization achievements in this study could be largely shared between these two models. NAQPMS is mainly used for regional high resolution (e.g. 1-10 km) air pollution simulation and routine air quality prediction. The typical time scale of these applications is several days or months. And the current computation performance of NAQPMS can generally meet the demand of these applications. GNAQPMS is designed for global-scale, long-term atmospheric aerosol and chemistry simulation, and it is also online coupled to the Earth System Model of Chinese Academy of Sciences (CAS-ESM) for study of climate change. These applications have typical time scale of more than 10 years, and raise very high requirements for model computation speed (e.g. 5-10 model years per day). Obviously, there is a large gap between the current computation performance of GNAQPMS and the actual need. Therefore, we choose GNAQPMS to start the code optimization.

A brief description of NAQPMS is added to Section 2.1 in the revised manuscript, and the reason why we focus on GNAQPMS is also given.

3. **Page 4, line 3: "Since the memory processor is not dominant". Unclear, explain further: what do the authors mean specifically by "memory pressure". Memory working set size? Memory bandwidth requirement of this application. How have they determined it is not dominant?**

**Reply:** Thanks for the comments. The memory bandwidth profiling given by VTune shows that the memory bandwidth requirement is far below the peak capability of both CPU and KNL platforms, as shown in Figure 2. In addition, the memory footprint of this workload is about 3G, which can be fully accommodated by the 16GB MCDRAM on KNL. In order to explain clearly, this sentence has been modified as following:

**"Since GNAQPMS is not memory bandwidth bounded, the cache mode is chosen in our experiment."**

4. **Page 4, line 9: Spell out first use of TLS to represent thread local storage. It is spelled out in the abstract, but needs to be spelled out again in main body of paper.**

**Reply:** Thanks for suggestion. TLS has been spelled out in the revised paper as following:

"…reducing **Thread Local Storage (TLS)** and changing the way of global communication in the GNAQPMS model."

5. **Page 5, paragraph beginning line 6 and all of Section 3.2.1. As noted in the general comments above, the use of files for global communication instead of MPI reductions and gathers is problematic for any modern parallel architecture, not just the KNL. Strongly recommend removing discussion of this optimization, including the entire Section 3.2.1, from the paper. It is relevant to optimizing for KNL.**

**Reply:** Thanks for the precious suggestion. This paper describe the common measures as well as the special measures for our model, and low frequency core in KNL lead to that I/O could have more effect on KNL compared with CPU (Talbe 1). To keep the integrity of our work, we suggest keeping the discussion in section 3.2.1 and separate the contribution of this part for speedup.

And because of the adjustment of the platform for single node testing, we retested the single

node results and showed in Table 1, including the results without the global communication. The results show that the optimal speedup on CPU and KNL are 2.77 and 3.50, respectively. Without the global communication optimization, the speedups drop to 2.69 (CPU) and 3.03 (KNL), respectively (Table 1).

**6.    Page 6, line 11. "Cyclic order"? Do the authors mean loop nesting order?**

**Reply:** Thanks for the comments. Yes, it means the loop nesting order. The sentence was revised as following:

"We changed **the nesting order** of loops from **j**, **i**, **igas** to **igas, j, i**···"

**7.    Page 6, line 12. "We cancelled the calling of the subroutine ... and made it an internal function in main program." This technique is referred to as inlining. Did the authors look at inlining reports from the Intel compiler to see if this could have been accomplished automatically or with the help of directives and compiler options, without manually restructuring the code? Also, the authors mean to say the "calling subroutine", not the "main program".**

**Reply:** Thanks for the comments. Yes, we've tried the "-ipo" option for automatic inlining and checked the compiler report. The report on this subroutine said "Inlining would exceed -inline-max-size value", which means inlining could not be done by the compiler due to unsatisfied heuristic. Therefore, we have to manually restructure the code at this step, and the calling site happens to be in the main program. According to the reviewer's comments, we modified the sentence as following:

**"At the second step, since the subroutine get_ratio_emit() is too big to be inlined automatically by the Intel compiler, we manually inline it in the calling site of main program, to improve the calling efficiency and facilitate the vectorization."**

**8.    Page 6, line 14. "... using parameters to convert scalar structure to vector structure." Unclear what "coverting scalar to vector" means, nor does it appear from step 3 Figure 3 that this is what is going on.**

**Reply:** Thanks for the comments. As the reviewer mentioned, this part means the step 3 in "*Fig.*

*3"* in the manuscript. In this part, we use the parameter arrays to construct the loop to do vectorization calculation. Therefore, we modified this sentence to explain it more clearly:

"**Thirdly, constructive vectorization was involved in the emission section of the model, using the parameter arrays to convert the assignments from scalar structure to loop structure, which got vectorized by the compiler finally, as showed by step three in Fig. 3.**"

9. **Page 6, line 15. "... we added the directives, clauses, declarations and syntax comment of OpenMP outside the outermost loop as shown in box 4." This sentence appears to be the only discussion in the paper of how and why OpenMP threading was added to the code. Given that this is one of the five optimizations being presented, this is insufficient. It's not clear why the authors felt it was important to add OpenMP threading nor whether, from the results, it provided a benefit.**

**Reply:** Thanks for the comments. As described in the response for the general comments at the beginning, our concern is mainly about two aspects. On the one hand, our goal is to accelerate the model and improve the scalability, and our primary desire is to replace the MPI processes by the relative cheap OpenMP threads as many as possible on each calculation node, which is also adopted by other large-scale models, such as CESM. On the other hand, considering the features of multi-core and low core frequency of KNL, the large number of pure MPI processes would lead to expensive communication cost for KNL and it could be inevitable to do the hybrid optimization. Currently, our OpenMP optimization did not achieve the primary goals, and the optimal number of threads for KNL and CPU are 6 and 4 (Table 1), respectively. As mentioned in the manuscript, the results we described is our first move and the OpenMP optimization is relatively simple. In the future, we will do more investigation into this part and rebuild the OpenMP code structure.

10. **Page 7, line 15. "2) updating some code segments to the the serial codes to construct vectorization." What does this mean?**

**Reply:** Thanks for the comments. It refers to construct loop to do the vectorization calculation, which is similar with the step 3 in *"Fig. 3"* in the manuscript. In order to avoid the confusion of this sentence, the original sentence is modified as following:

**"converting the scalar structure to loop structure for compiler vectorization, as showed in step 3 in Fig. 3, but using more complex parameter arrays to build the loop structure;"**.

11. **Page 7, lines 17-25. A reader might make a number of educated guesses about what the authors are saying was the bottleneck and how it was fixed, but it is not clear at all from the text. What codes were "added by the compiler?" Was the issue a copyin/copy-out problem for thread-local variables that were listed as private/firstprivate in the OpenMP directives? If so, how did adding the CBZOBJ argument fix this? By passing-by-reference? If so, how were data races avoided? Were they avoided because the CBMZOBJ objects themselves were THREADPRIVATE? There's a lot of important detail missing from this discussion.**

**Reply:** Thanks for the precious comments. Firstly, please allow us to clarify the performance issue with common variables in OpenMP TLS (Thread Local Storage). The TLS is introduced for variables in named common blocks when using threadprivate OpenMP directive, and allocated for each thread on thread creation. These variables are private to each thread and global within the thread. When a thread references a common variable in its TLS, the memory address of TLS is first located by calling an OpenMP library function with the thread ID, then the common variable is addressed within the TLS space. Even for the references to common variables within the same named common block in the same subroutine, the above process is repeated for every variable, rather than addressing the TLS and common block only once. Since calling the OpenMP library function for TLS addressing is expensive, and there are many references to these common variables in the user subroutines, the total overhead of using common variables in TLS is extremely high. Linking against static OpenMP library can alleviate the calling cost partially but the cost is still unbearable.

Basically, our solution is to construct a derived type (CBMZTYPE) object to eliminate the expensive OpenMP function calling. In the optimized code, the common variables are removed from named common blocks and added to the CBMZTYPE as its members. Using PIRVATE list in the OpenMP directive, each thread owns a private copy of object instance of CBMZTYPE, i.e. the cbmzobj variable. Since the cbmzobj is located on thread local stack, the references to its member variables require only simple relative addressing on stack, with simple yet efficient

instructions. Meanwhile, since the common variables in the original code are no longer global and visible within the user subroutines, a formal parameter (argument) of cbmzobj is added to the subroutines using the variables. The additional overhead of passing the address of cbmzobj to the subroutine is quite small. Therefore, the cost of referencing common variables in TLS is greatly reduced with the derived type object.

**Now we try to answer the reviewer's questions.**

**Q: What codes were "added by the compiler?"**

**A:** It means the codes of referencing the common variable in TLS, including the calling to OpenMP library function for TLS address, and then accessing the common variable within the TLS space.

**Q: Was the issue a copyin/copy-out problem for thread-local variables that were listed as private/firstprivate in the OpenMP directives?**

A: No, it is a reference cost issue, for variables in named common blocks that are listed in the THREADPRIVATE directive.

**Q: If so, how did adding the CBZOBJ argument fix this? By passing-by-reference? If so, how were data races avoided? Were they avoided because the CBMZOBJ objects themselves were THREADPRIVATE?**

**A:** The issue is not fixed by adding the CBMZOBJ argument, but by constructing the CBMZOBJ object to store the variables in the original common blocks. Since the CBMZOBJ object is thread private and is on the thread local stack, the references to its members are quite cost efficient.

12. **Page 7, line 32. "...which spent 10 percentages and 8 percentages." One would prefer to see actual timings instead of percentage of runtime in discussion of performance improvement.**

**Reply:** Thanks for the comments. Firstly, we updated the data in Fig. 2 because the previous results came from an old test. And the description of Fig.2 in the manuscript is revised accordingly.

The aim of this part is to emphasize the role of different sections of the time consumption. The impact of the optimization on these sections is showed in Line 3, Page 8, with the speedup**.**

Upon the comments of the reviewer, we added the walltime as well as the speedup to illustrate the improvement:

"…**According the performance on the single node, the diffusion module can get 1.99X speedup (from 241.97s to 121.19s) on the CPU platform and 3.31X speedup (from 241.97s to 73.05s) on the KNL platform**"

"…**Finally, the optimized wet deposition module got 5.91X speedup (from 498.01s to 84.13s) on the KNL platform, much higher than 3.18X speedup (from 498.01s to 156.13s) on the CPU platform.**"

13. **Page 7, line 36. "1.78 speedup on the CPU platform and 2.39 speedup on the KNL." How much of these speedups were from the manual vectorization and how much from the optimization of global communication? (Again, this reviewer suggests not considering global communication optimization at all, but if it is discussed, show effects separately from other optimizations).**

**Reply:** Thanks for the comments. As mentioned in the response for specific comment 5 above, we tend to discuss the global communication as part of our work, and we have separated the contribution of this part from the total speedup as required by the reviewer. The optimal speedup on CPU and KNL are 2.77 and 3.50, respectively. Without the global communication optimization, the speedups of same combination of OpenMP and MPI will drop to 2.69 (CPU) and 3.03 (KNL), respectively (Table 1).

14. **Page 8, lines 7-10. "The horizontal resolution of the model is 1°×1°, which indicates that the modelling domain contains 360×180 grids. And the number of vertical layers is 20, while the time step for integration is 600 seconds in the test case. The test case was designed to test the performance of GNAQPMS on single node of CPU and KNL platform, and multi-nodes on different platform clusters." This workload is very small and probably not suitable for KNL clusters, which require a high degree of parallelism to be efficient. (The authors make note of this on page 9, lines 32-33). The paper describe a representative workload for scientific simulations using the GNAQPMS and provide some discussion of how the performance results presented**

**are relevant to an actual scientific workload.**

**Reply:** Thanks for the comments. We generally agree with the reviewer. Currently, due to limited HPC resources and poor model computation performance, the typical resolution for global aerosol and chemistry simulation in scientific research is about 2°x2°. It makes sure that a several year's model simulation can be accomplished in a month. The test case in this study was chosen based on the following reasons: 1) The configuration with 1°x1° horizontal resolution and 600 s integration time step is common in scientific applications. This model configuration has also been used in several previous studies (Chen et al., 2015). By using this configuration, the achievements of this study can be directly applied to actual scientific applications. 2) The test case is a medium-scale workload for global chemistry simulation, which allows us to carry out a lot of debugging and testing. 3) This is the first time to port and optimize GNAQPMS on the KNL platform. A lot of fundamental work is needed to optimize the code and solve potential bugs. Therefore, choosing a medium-scale test case would be a good start.

We agree that a large workload with high model resolution is probably more suitable for KNL clusters, and it might get better parallel scalability. In addition, based on our tests, the bottleneck of MPI global communication and fragmental OpenMP parallel regions are also main reasons for the poor parallel scalability of GNAQPMS on KNL clusters. Global high resolution simulation is a clear trend for the development of chemical transport models. Testing and optimizing GNAQPMS with a super large workload (e.g. 0.25°x0.25° or 0.1°x0.1°) will be the next emphasis in the near future.

Further description of the test case is added to Section 4 (the first paragraph) in the revised manuscript.

15. **Page 8, line 15. "The Intel Corporation provides the High Performance Computing environment for the test." Was this Intel's Endeavor cluster? Perhaps mention this as well.**

**Reply:** Thanks for the comments. There are two sets of platforms were used. The single node tests were conducted on **Cthor Lab.** and the cluster tests were done on **Intel Endeavor** cluster. Considering the stability of the test environment, we adopted the machines in **Cthor Lab.** for

the single-node test. According to the suggestion, we have mentioned the names of test environment in the revised manuscript:

"**The Intel Corporation provides the High Performance Computing environment (Cthor Lab. for single node tests and Endeavor Cluster for cluster tests).**"

16. **Page 8, line 20. Opt-V GNAQPMS has been compiled on CPU and KNL platform, respectively." Not sure what this means. Are the authors actually compiling the codes on the respective platforms? If so, why? Is there some difference expected between a native-compiled and a cross-compiled executable?**

**Reply:** Thanks for the comments. Both native and cross compiling work, and there is no difference between them. Actually, we compiled the same codes with different compiler flags showed in Table 2 on the same CPU platform. In order to avoid such confusion, we modified this part as following:

"…and the Opt-V GNAQPMS has been compiled **for** CPU and KNL platform with the compiler flags showed in Table 2**.**"

17. **Page 8, line 21. "... the -xCore-AVX2 and –xMIC-AVX512 compile flags were not used for the advection module..." This is troubling and bears further discussion. What did the authors see that caused them to avoid these compiler flags for the advection module? Why is advection susceptible to differences but not other parts of the code such as the ODE solver?**

**Reply:** According to this comment, we can also answered part of comment 18 at the same time. Actually, we designed two mechanisms to verify our results from the two models (Opt-V, Base-V). At first, we added an extra module to test our results. The function of this module is outputting the concentration of specific chemical species after each chemical or physical processes. Each process writes its own data into its own files respectively at the same time, and each chemical and physical module would only run one time to insulate the effect of other modules. Then, an additional small program will read the output files from the two version of GNAQPMS to check and report the relative and absolute errors. This method is likes that sampling the results during the running period. In this way, we can find the main error between

each sections. On the other hand, we could plot the spatial distribution of the two model results after a long time integration, as we did in the paper. And according to the first step, after a serious of test and debugging, we found that the compile flag, such as -xCore-AVX2 and –xMIC-AVX512, would affect the results because of the sensitiveness of calculation accuracy. Although the same –fp-model flag would reduce the error raised by advection, the error could not be completely diminished because of numerical sensitivity of advection algorithm.

In addition, we did not find the obvious errors caused by the ODE solver in current version codes by the previous methods, since the ODE solver is not as sensitive as the advection section. Generally, there is no obvious difference between the spatial distribution by calculating the Relative Error (Figure 4) and Relative Mean Square Difference (Figure 5), and the white part in the RE columns indicates the error is small enough (<1%).

[Figure]

**Figure 5. The Relative Error between the results of Base-V and Opt-V GNAQPMS.**

[Figure]

**Figure 6. The Relative Mean Square Difference between the results of Base-V and Opt-V GNAQPMS.**

18. **Page 9, line 2. "...were confirmed to be identical." This is not a persuasive verification method. An eyeball comparison of plots would not be considered sufficient to confirm that results are identical. Provide difference plots and RMS difference statistics. Page 9, line 20. "...when the computing scale is fixed." Right word? Instead maybe workload? problem size?**

**Reply:** Thanks for your kind comments. The answer for validation of model results has been given in the above section, Question 17.

According to the suggestion, the Page9, line 20 is modified to explain more suitable:

"**...when the workload is fixed**".

19. **Page 9, line 24. "After optimization, the parallel scalability of GNAQPMS is greatly improved..." This seems counterintuitive, since vectorization and other node-performance optimizations that improve performance on each node from should make the code less well, assuming that interprocessor communication overheads are**

**the same. It's important to distinguish here (1) what parts of the code are being timed for the scaling measurements. (2) Which optimizations are contributing to the improved scalability and which may be improving performance but working against strong scaling.**

**Reply:** Thanks for your comments. As mentioned in Section 3.2, the optimizations include global communication optimization that replaces interface-files writing/reading with collective MPI functions. This optimization reduces the communication overhead greatly, since the overhead of file I/O increases much faster than that of collective MPI functions as the number of nodes increases. For the optimized code, since the communication improvement is much more than that of the computation part, we observe a great improvement in parallel scalability. To (1) question, we evaluate the run time for "the core calculation portion of the model", as mentioned in Page 9, lin23.

To (2) question, with respect to reviewer's classification, we would declare that the global communication optimization contributes to the scalability improvement, and other optimizations mainly contribute to the computation improvement. But we would like to further point out that single-node optimization does not necessarily work against strong scaling. There are at least two exceptions. One is the OpenMP parallelization that could hide the latency of MPI point-to-point communication partially or fully. In some cases, it can help improve the single-node performance as well as the cluster scalability. Another is the CA-KSMs (Communication-Avoiding Krylov Subspace Methods) developed by Professor Demmel at Berkeley, at algorithmic level, which optimize the global communication and expose the opportunity for local computation optimization at the same time.

20. **Page 9, line 33. "... OpenMP code segments on KNL..." What is the effect of varying the number of OpenMP threads with respect to MPI tasks? That is, using the same overall number of hardware threads across the job? What is the effect of running pure MPI? Pure OpenMP?**

**Reply:** Thanks for your kind comments. Firstly, we unfortunately found that single node test environment (Cthor Lab.) had some adjustment of the machines, which lead to the difference of the test results. Therefore, we retested all the results and updated the data in the revised

manuscript. As suggested by the reviewer, we tested the combination of different OpenMP threads and MPI processes (Table 1). All tests are fully using the hardware threads. The results indicate that the best combination on CPU platform is 6 OpenMP with 12 MPI processors. For KNL, we used the command line "–env I_PIN_MPI_DOMAIN=N" to pin the MPI processes to specific cores. Moreover, the combination of 4 OpenMP and 34 MPI processes could get the optimal speedup (3.509, which is 3.34 in the original manuscript).

**Table 1 the speedup and walltime of different combination of OpenMP threads and MPI processes.**

| CPU(E5-2697 V4 with 36 physical cores and 2 hyperthreads) | | | | |
|---|---|---|---|---|
| | OMP | MPI | WALLTIME | SPEEDUP |
| Baseline (No HyperThread) | 0 | 36 | 4381.2 | 1 |
| Opt-V | 1 | 72 | 1769 | 2.477 |
| | 2 | 36 | 1625.72 | 2.695 |
| | 4 | 18 | 1614.9 | 2.713 |
| | 6 | 12 | 1580.1 | 2.773 |
| | 12 | 6 | 1612.3 | 2.717 |
| | 18 | 4 | 1790.2 | 2.447 |
| | 36 | 2 | 2243.4 | 1.952 |
| Opt-V(No global communication) | 6 | 12 | 1623.6 | 2.698 |
| KNL(KNL 7250 with 68 physical cores and 4 threads) | | | | |
| Opt-V | 2 | 136 | 1499.2 | 2.922 |
| | 4 | 68 | 1402.9 | 3.12 |
| | 2 | 68 | 1512.8 | 2.896 |
| | 4 | 34 | 1248.3 | 3.509 |
| | 8 | 34 | 1373.6 | 3.189 |
| | 16 | 17 | 1473.2 | 2.974 |

| Opt-V(No global communication) | 4 | 34 | 1444.6 | 3.032 |
| --- | --- | --- | --- | --- |

**21. Page 10, Conclusion section. Future work and areas for improvement are discussed. However the conclusion should also include discussion of what levels of performance and scaling are needed for simulation science using GNAQPMS. The conclusion should assess whether and how well the presented work helps the GNAQPMS users achieve these goals.**

**Reply:** Thanks for the comments. We agree with the reviewer. The levels of performance and scaling needed for simulation science using GNAQPMS has been discussed in the response of Question One. For short-term or medium-term simulations (5 years or less), the model computation speed needed is about 1 model year per day (including model I/O), while it should be increased to about 5 model years per day for long-term (30 years or more) high resolution simulations (e.g. 0.25°x0.25°). Improve single node computation speed and parallel scalability is the way to achieve the above goals. As shown in the conclusion, the single node computation speed and parallel scalability of GNAQPMS were significantly improved after code optimization. The computation speed (excluding model I/O) has been improved from about 0.5 model year per day using 8 CPU nodes to about 3.7 model year per day using 30 KNL nodes and about 8 model year per day using 40 CPU nodes, respectively. Therefore, without regard to the model I/O, the optimization work in this study has made the computation performance of GNAQPMS very close to our anticipated goal. In the next step, further work will be focused on solving problems concerning OpenMP parallel regions and global communication on KNL platform and conducting analysis and optimization of model I/O.

Further discussion of the model computation performance needed now and in the future is added to Section 5 (the second paragraph) in the revised manuscript:

**"In summary, the computation speed (excluding model I/O) has been improved from about 0.5 model year per day using 8 CPU nodes to about 3.7 model year per day using 30 KNL nodes and about 8 model year per day using 40 CPU nodes, respectively. Therefore, without regard to the model I/O, the optimization work in this study has made the computation performance of GNAQPMS very close to our anticipated goal. In the next step, further work will be focused on merging OpenMP parallel regions."**

**22. Figure 7. Shows speedup but not performance. Suggest adding a figure that shows performance as a function of the number of nodes. Plot as simulation seconds per second of wall clock time.**

**Reply**: Thanks for your kind comments. We added the following figure that shows the performance as a function of the number of nodes, and this figure will be added into the supplementary material:

[Figure]

**Figure 7 the performance ability with increasing of number of nodes**

**References:**

Chen, H. S., Wang, Z. F., Li, J., Tang, X., Ge, B. Z., Wu, X. L., Wild, O., and Carmichael, G. R.: GNAQPMS-Hg v1.0, a global nested atmospheric mercury transport model: model description, evaluation and application to trans-boundary transport of Chinese anthropogenic emissions, Geosci. Model. Dev., 8, 2857-2876, doi:10.5194/gmd-8-2857-2015, 2015.

Li, J., Wang, Z., Akimoto, H., Gao, C., Pochanart, P., and Wang, X.: Modeling study of ozone seasonal cycle in lower troposphere over east Asia, J. Geophys. Res.-Atmos., 112, D22s25, doi:10.1029/2006jd008209, 2007.

Li, J., Wang, Z., Wang, X., Yamaji, K., Takigawa, M., Kanaya, Y., Pochanart, P., Liu, Y., Irie, H., Hu, B., Tanimoto, H., and Akimoto, H.: Impacts of aerosols on summertime tropospheric

photolysis frequencies and photochemistry over Central Eastern China, Atmos. Environ., 45, 1817-1829, doi:10.1016/j.atmosenv.2011.01.016, 2011.

Li, J., Wang, Z., Zhuang, G., Luo, G., Sun, Y., and Wang, Q.: Mixing of Asian mineral dust with anthropogenic pollutants over East Asia: a model case study of a super-duststorm in March 2010, Atmos. Chem. Phys., 12, 7591-7607, doi:10.5194/acp-12-7591-2012, 2012.

Tang, X., Wang, Z., Zhu, J., Gbaguidi, A. E., Wu, Q., Li, J., and Zhu, T.: Sensitivity of ozone to precursor emissions in urban Beijing with a Monte Carlo scheme, Atmos. Environ., 44, 3833-3842, doi:http://dx.doi.org/10.1016/j.atmosenv.2010.06.026, 2010.

Wang, Q., Fu, Q., Wang, Z., Wang, T., Liu, P., Lu, T., Duan, Y., and Huang, Y.: Application of ensemble numerical model system on the air quality forecast in Shanghai (in Chinese), Environmental Monitoring and Forewarning, 2(4), 1-6+11, 2010.

Wang, Z., Akimoto, H., and Uno, I.: Neutralization of soil aerosol and its impact on the distribution of acid rain over east Asia: Observations and model results, J. Geophys. Res.-Atmos., 107, 4389, doi:10.1029/2001jd001040, 2002.

Wang, Z., Xie, F., Wang, X., An, J., and Zhu, J.: Development and application of Nested Air Quality Prediction Modeling System (in Chinese), Chinese Journal of Atmospheric Sciences, 30(5), 778-790, 2006.

Wang, Z., Wu, Q., Gbaguidi, A., Yan, P., Zhang, W., Wang, W., and Tang, X.: Ensemble air quality multi-model forecast system for Beijing (EMS-Beijing): Model description and preliminary application (in Chinese), Journal of Nanjing University of Information Science & Technology (Natural Science Edition), 1(1), 19-26, 2009.

Wu, Q., Wang, Z., Chen, H., Zhou, W., and Wenig, M.: An evaluation of air quality modeling over the Pearl River Delta during November 2006, Meteorol. Atmos. Phys., 116, 113-132, doi:10.1007/s00703-011-0179-z, 2012.

---

## Author Comment (AC2) · 18 May 2017

Dear reviewer,

Thanks a million for your precious time and comments. We will reply the reviewer's comment point-to-point in the following part.

"**The paper describes the work done by the authors on optimizing the GNAQPMS model for Intel Knights Landing. The paper is well organized. I agree with the authors that there are still options to further optimize the code but the work performed so far isa good first step and worth publishing. There are a few changes that need to be addressed. Also the language could use some general improvement. I will first give some specific comments that I believe need to be addressed before publication. After that I list a few (optional) suggestions for future optimizations.**"

**Reply:** We really appreciate for your praise. We believe this work could be a good example for the model developer who wants to transplant their model to KNL platform. Thanks for your comments. As for the language and grammatical issues, we will invite the native English speaker or some relative company to do the copy editing for English.

According to your specific comments, the following modifications have been done.

(a) **section 1, page 3: according to the introduction Section 3.1 is supposed to discuss where the bottlenecks of the code are. I assume that this refers to the runtime measurements shown in Figure 2? Under the term "bottleneck" I would have expected a discussion whether the code is bound by memory bandwidth by showing measurements of memory bandwidth and flops/s and comparing them with the peak performance obtained for the STREAM and LINPACK benchmarks. I understand that hardware counters are not very accurate on Intel architectures but you could still count them with an emulator or by hand. The paper does not show in its current state what the bottlenecks of the different parts of the code are.**

**Reply:** Thanks for your kind comments. We agreed with the reviewer. We have tried to measure the memory bandwidth and flops/s of the Base-V GANQPMS by the Vtune tools (R1).

**R1 the HPC-performance report of the Base-V GNAQPMS**

The FLOP/S of the Base-V GNAQPMS detected by Vtune showed in R1 is about 93.714 GFLOPS, and that of the Opt-V GNAQPMS is reaching 279.326 GFLOPS (R2). The Vtune Memory Bound measures the fraction of slots where pipeline could be stalled due to demand load or store instructions. And both of two reports indicate that the memory bound is not the dominant limitation of model performance.

**R2 the HPC-performance report of Opt-V GNAQPMS**

However, because of the bad modularity of GNAQPMS, we cannot get the testing data of GFLOPS and bandwidth of every section of the model, and data we got is based on the function- and loop-level. And obtaining the data of each section may lead to a large amount of work to pack the code, which could not be finished immediately. According to the current test results, we still can draw a preliminary conclusion that the insufficient use of vectorization is still a main bottleneck for our model but not the bandwidth. And we have done the further work to improve the vectorization and got a good speedup, which would be presented in the future paper.

**(b) section 4.2, page 9: comparing the patterns and values without giving any specific relative difference between base and optimized version of the code is not enough to claim that the results are identical. The authors need to show some precise numbers like the total mass of the air and the different chemical species in both versions and the difference between optimized and base version.**

**Reply:** Thanks for your precious comments. Actually, we designed two mechanisms to verify our results from the two models (Opt-V, Base-V). At first, we added an extra module to test

our results. The function of this module is outputting the concentration of specific chemical species after each chemical or physical processes. The processes write its own data into its own files respectively at the same time, and each chemical and physical module would only run one time to insulate the effect of other modules. Then, an additional small program will read the files from the two version of GNAQPMS and calculate/report relative error and absolute error. This method is likes that sampling the results during the running period. By this way, we can find the primary error due to every section. On the other hand, we could plot the spatial distribution of two model results after a long time integration, as we did in the paper. Actually, we calculated the difference of two plots in Fig. 5(in paper), and considering the beauty, we only put the two spatial distribution plots. And according to the first step, after a serious of test and debugging, we found that the compile flag, such as -xCore-AVX2 and –xMIC-AVX512, would affect the results because of the sensitiveness of calculation accuracy. The same –fp-model flag could reduce the error raised by advection, but the error could not be completely diminished because of numerical sensitivity.

Generally, there is no obvious difference between the spatial distribution by calculating the Relative Error (R3) and Relative Mean Square Difference (R4), and the REs of almost all grids are lower 1%, which, we think, is acceptable.

[Figure]

**R3 he Relative Error between the results of Base-V and Opt-V GNAQPMS.**

[Figure]

**R4 The Relative Mean Square Difference between the results of Base-V and Opt-V GNAQPMS.**

**(c) section 4.3, page 9: how did you measure the power consumption? Do you trust VTune to give you these measurements or did you measure the power consumption yourself?**

**Reply:** Thanks for your precious comments. Both Xeon and Xeon Phi platforms have the same built-in power and thermal sensors. We measured the power consumption with an IPMI-based script tool to query these sensors from a remote server at a constant interval, such as 0.02 seconds. IPMI, short for "Intelligent Platform Management Interface", is a low-level interface specification that allows remote management at the hardware level without dependencies on the operating system. IPMI communicates with the server baseboard management controller (BMC), which is a reliable agent in the system for management and gathering system health, including power consumption data.

**(d) How many OpenMP threads were used per MPI process? Did you try different configurations (like 2, 4, 8, 16, 32, 64 threads per MPI process)?**

**Reply:** Thanks for your kind comments. At first, we unfortunately found that single node test environment (Cthor Lab.) had some adjustment of the machines, which lead to the difference of test results. Considering the time saving, our previous tests about the combination of

OpenMP threads and MPI processes are implemented by using the 1h-running test; we chose the best combination to do the 48h-test. Therefore, we retested all the results by testing the 48h workload and updated the data in the revised paper (Table 1). We tested the combination of different OpenMP threads and MPI processes as suggested by the reviewer. All of tests are fully using the hardware threads. And the results indicates that the best combination on CPU platform is 6 OpenMP with 12 MPI processors. For KNL, we use the command line "–env I_PIN_MPI_DOMAIN=N" to pin the OpenMP threads to the MPI processes. Moreover, the combination of 4 OpenMP and 34 MPI processes could get the optimal speedup (3.51, which is 3.34 in discussion paper).

**Table 1 the speedup and walltime of different combination of OpenMP threads and MPI processes.**

| CPU (E5-2697 V4 with 36 physical cores and 2 hyperthreads) | | | | |
|---|---|---|---|---|
| | OMP | MPI | WALLTIME | SPEEDUP |
| Baseline (No HyperThread) | 0 | 36 | 4381.2 | 1 |
| Opt-V | 1 | 72 | 1769 | 2.477 |
| | 2 | 36 | 1625.72 | 2.695 |
| | 4 | 18 | 1614.9 | 2.713 |
| | 6 | 12 | 1580.1 | 2.773 |
| | 12 | 6 | 1612.3 | 2.717 |
| | 18 | 4 | 1790.2 | 2.447 |
| | 36 | 2 | 2243.4 | 1.952 |
| KNL(KNL 7250 with 68 physical cores and 4 threads) | | | | |
| Opt-V | 2 | 136 | 1499.2 | 2.922 |
| | 4 | 68 | 1402.9 | 3.12 |
| | 2 | 68 | 1512.8 | 2.896 |
| | 4 | 34 | 1248.3 | 3.509 |
| | 8 | 34 | 1373.6 | 3.189 |
| | 16 | 17 | 1473.2 | 2.974 |

**(e) The term "manual vectorization" is used many times throughout the paper. This is very misleading. Manual vectorization would be in my opinion if the code was rewritten with avx512 vector intrinsics in C! The vectorization used in this paper still relies on the compiler.**

**Reply:** Thanks for the comment. We agreed with the reviewer. To avoid misleading the readers, we consider that "vectorization with compiler's directives" may be a more accurate expression. And we will update this part in the revised paper.

**Suggestions**

**(f) As mentioned before the paper does not investigate what the real bottleneck of the code is and how far the code is from optimal performance. I highly recommend to create a theoretical and measured roofline model. This would allow to answer how much potential for further optimization should still be possible.**

**Reply:** Thanks for the precious comments. As we discussed in the Question 1, it is not easy to investigate the rootline of every section, and we can provide the general report in R1 and R2 now. We think it could need more time to test and establish a reliable roofline model. According to the test results, we consider that the model still has the optimization potential. And our goal is that our model can get the speed of 5-10 model/day.

**(g) I would love to know how many of all floating point operations are vectorized. I understand that counting floating point operations on Intel architectures through hardware counters is not very accurate. Maybe you could give some estimates from what the vectorization report shows you?**

**Reply:** Thanks for your kind suggestion. From R1 and R2, the FPU utilization is increased from 2.9% to 9.6% after optimization, and the vector capacity usage is increased from 24% to 65%.

---

## Author Response (AR1)

We thank the reviewers for constructive comments and suggestions. We have revised the manuscript accordingly. The tile of our manuscript has been updated with model version as "**GNAQPMS v1.1: Accelerating the Global Nested Air Quality Prediction Modeling System (GNAQPMS) on Intel Xeon Phi Processors**"

In the following, we provide a detailed point-by-point response and specify all changes in the revised manuscripts

**Response to Referee #1:**

**Response to the General Comments of Referee #1:**

1) **"The paper describes the work done by the authors on optimizing the GNAQPMS model for Intel Knights Landing. The paper is well organized. I agree with the authors that there are still options to further optimize the code but the work performed so far is a good first step and worth publishing. There are a few changes that need to be addressed. Also the language could use some general improvement. I will first give some specific comments that I believe need to be addressed before publication. After that I list a few (optional) suggestions for future optimizations."**

**Reply:** We really appreciate for your praise. We believe this work could be a good example for other model developers who want to transplant their models to KNL platform. Thanks for your comments. As for the language and grammatical issues, we have invited the language service company **Elsevier** to do the copy editing for English. The certificate is presented as the following:

[Figure]

**Language Editing Services**

*Registered Office:*
Elsevier Ltd
The Boulevard, Langford Lane,
Kidlington, OX5 1GB, UK.
Registration No. 331566771

**To whom it may concern**

The paper "GNAQPMS v1.1: Accelerating the Global Nested Air Quality Prediction Modeling System (GNAQPMS) model on Intel Xeon Phi processors" by Hui Wang, Huansheng Chen, Qizhong Wu, Junmin Lin, Xueshun Chen, Xinwei Xie, Rongrong Wang, Xiao Tang, Zifa Wang was edited by Elsevier Language Editing Services.

Kind regards,

Biji Mathilakath
**Elsevier Webshop Support**

**Response to the Specific Comments of Referee #1:**

2)  **section 1, page 3: according to the introduction Section 3.1 is supposed to discuss where the bottlenecks of the code are. I assume that this refers to the runtime measurements shown in Figure 2? Under the term "bottleneck" I would have expected a discussion**

**whether the code is bound by memory bandwidth by showing measurements of memory bandwidth and flops/s and comparing them with the peak performance obtained for the STREAM and LINPACK benchmarks. I understand that hardware counters are not very accurate on Intel architectures but you could still count them with an emulator or by hand. The paper does not show in its current state what the bottlenecks of the different parts of the code are.**

**Reply:** Thanks for your kind comments. We agreed with the reviewer. We have tried to measure the memory bandwidth and flops/s of the Base-V GANQPMS by the Vtune tools.

[Figure]

**Figure R1. The HPC-performance report of the Base-V GNAQPMS**

The FLOP/S of the Base-V GNAQPMS detected by Vtune showed in Figure R1 is about 93.741 GFLOPS, and that of the Opt-V GNAQPMS is reaching 279.479 GFLOPS showed in Figure R2. The Vtune Memory Bound measures the fraction of slots where pipeline could be stalled due to demand load or store instructions. Both two reports indicate that memory bound is not the dominant limitation of model performance.

[Figure]

**Figure R2. The HPC-performance report of the Opt-V GNAQPMS**

However, because of bad modularity of GNAQPMS, we couldn't get the testing data of GFLOPS and bandwidth of every section of the model code. Obtaining the performance of each section may lead to a large amount of work to pack the code, which could not be done immediately. According to the current testing results, we still can draw a preliminary conclusion that the insufficient use of vectorisation is a main bottleneck of our model but not the bandwidth.

According to the comments, a new table, Table 1, with the HPC reports from Vtune was added to the revised manuscript, and the following description was also added in Sec. 3.1 Paragraph 1:

**"By using the VTune HPC performance detection tool, we could report the general performance, e.g. GFLOPS, bandwidth, CPU and FPU utilisation, through a simple report. Table 1 presents the general indicators detected by the VTune HPC performance detection tool for the two models. Moreover, an obvious increase in GFLOPS was detected from 93.741 to 279.479. The Memory Bound in Table 1 indicates the fraction of slots where a pipeline could be stalled owing to the demand load or store instruction, and the values of 9.2% of Base-V and 12.7% of Opt-V indicate that the bandwidth is not the limitation of our model. The FPU utilisation was also improved from 2.9% to 9.6%, although there is still room for improvement.**

**Further analysis of the hotspots and bandwidth could be detected by Hotspot and Memory-Access in VTune, respectively."**

3) **Section 4.2, page 9: comparing the patterns and values without giving any specific relative difference between base and optimized version of the code is not enough to claim that the results are identical. The authors need to show some precise numbers like the total mass of the air and the different chemical species in both versions and the difference between optimized and base version.**

**Reply:** Thanks for your precious comments. Actually, we designed two mechanisms to verify our results from the two model versions (Opt-V, Base-V). At first, we added an extra module into GNAQPMS to test our results. The function of this module is to output the concentration of specific chemical species after each chemical and physical processes. The MPI processes write their own data into their own files respectively at the same time, and each chemical and physical modules would only run one time to insulate the effect of other modules. Then, an additional program reads the files from the two version of GNAQPMS and calculates the relative and absolute errors. This method is similar with sampling the results during the running period. By this way, we can find the major errors due to every sections. On the other hand, we plotted the spatial distribution of modeling results from the two models after a long time integration, as we did in the manuscript. Actually, we have calculated the difference of the two plots in Figure 5 (in the manuscript). But considering the beauty, the spatial distribution of the difference between the two modeling results were not provided. According to the first step, after a series of test and debugging, we found that the compile flag, such as -xCore-AVX2 and –xMIC-AVX512, would affect the results because of the sensitiveness of calculation accuracy. The –fp-model flag could reduce the error raised by advection, but the error could not be completely diminished because of numerical sensitivity.

Generally, there is no obvious difference between the spatial distribution by calculating the Relative Errors in Figure R3 and Relative Mean Square Errors in Figure R4. The REs of almost all grids are lower than 1%, and we think this is acceptable.

In the revised manuscript, we have replaced Figure 5 with Figure R3 in this document to show the differences between the two modeling results, and the following Paragraph has also been added in Sec. 4.2:

"By comparing the model output results and plotting the spatial distribution images with the relative error (RE) shown in Figure 5, we can see, in the third column, that the RE was small (<1%) enough. The optimisation does not introduce an "erroneous" concentration for any atmospheric specie, and, therefore, it is reliable. However, the error could not be completely diminished because of the numerical sensitivity of the advection algorithm."

[Figure]

**Figure R3. The Relative Errors between the results of the Base-V and Opt-V GNAQPMS.**

[Figure]

**Figure R4. The Relative Mean Square Differences between the results of the Base-V and Opt-V GNAQPMS.**

4) Section 4.3, page 9: how did you measure the power consumption? Do you trust VTune to

**give you these measurements or did you measure the power consumption yourself?**

**Reply:** Thanks for your precious comments. Both Xeon and Xeon Phi platforms have the same built-in power and thermal sensors. We measured the power consumption with an Intelligent Platform Management Interface (IPMI)-based script tool to query these sensors from a remote server at a constant interval, such as 0.02 seconds. IPMI is a low-level interface specification that allows remote management at the hardware level without dependencies on the operating system. IPMI communicates with the server baseboard management controller (BMC), which is a reliable agent in the system for management and gathering system health, including power consumption data.

In the revised manuscript, the description of how to measure the power consumption is added in Sec.4.3 as follows:

**"The power consumption was measured with a script tool based on the Intelligent Platform Management Interface (IPMI). IPMI is a low-level interface specification that allows remote management at the hardware level without dependencies on the operating system. IPMI communicates with the server's baseboard management controller (BMC), which is a reliable agent in the system for managing and gathering system health, including power consumption data."**

5) **How many OpenMP threads were used per MPI process? Did you try different configurations (like 2, 4, 8, 16, 32, 64 threads per MPI process)?**

**Reply:** Thanks for your kind comments. At first, we unfortunately found that single node test environment (**Cthor Lab.**) had some adjustment of the machines, which lead to the difference of the test results. To save time, our previous tests about the combination of OpenMP threads and MPI processes were implemented by using the 1h-running test, and we chose the best combination to do the 48h-running test. Therefore, we retested all the results using the 48h workload and updated the data in the revised manuscript (Table 2 in the revised manuscript). We tested the combination of different OpenMP threads and MPI processes as suggested by the reviewer (as shown in Table R1). All the tests are fully using the hardware threads. The results indicates that the best combination on CPU platform is 6 OpenMP with 12 MPI processes. For KNL, we used the command line "–env I_PIN_MPI_DOMAIN=N" to pin the OpenMP threads to the MPI processes. And the combination of 4 OpenMP and 34 MPI processes could get the optimal speedup (3.51, which is 3.34 in the

original manuscript).

In the revised manuscript, we added **Table R1** as **Table 4**, and removed the previous Figure 6. The corresponding description was also added in Sec. 4.3 as follows:

**"Different combinations of OpenMP and MPI processes were tested on a single node, and the results are shown in Table 4. The speedup of the best combination for Opt-V GNAQPMS reached 3.51× on KNL and 2.77× on the CPU, compared to that of Base-V GNAQPMS, and the KNL platform had an advantageous speedup of 1.26× over the CPU platform."**

**Table R1. The speedup and walltime of different combination of OpenMP threads and MPI processes.**

| CPU (E5-2697 V4 with 36 physical cores and 2 hyperthreads) | | | | |
|---|---|---|---|---|
| | OMP | MPI | WALLTIME | SPEEDUP |
| Baseline (No HyperThread) | 0 | 36 | 4381.2 | 1 |
| Opt-V | 1 | 72 | 1769 | 2.477 |
| | 2 | 36 | 1625.72 | 2.695 |
| | 4 | 18 | 1614.9 | 2.713 |
| | 6 | 12 | 1580.1 | 2.773 |
| | 12 | 6 | 1612.3 | 2.717 |
| | 18 | 4 | 1790.2 | 2.447 |
| | 36 | 2 | 2243.4 | 1.952 |
| Opt-V(No global communication) | 6 | 12 | 1623.6 | 2.698 |
| KNL(KNL 7250 with 68 physical cores and 4 threads) | | | | |
| Opt-V | 2 | 136 | 1499.2 | 2.922 |
| | 4 | 68 | 1402.9 | 3.12 |
| | 2 | 68 | 1512.8 | 2.896 |
| | 4 | 34 | 1248.3 | 3.509 |
| | 8 | 34 | 1373.6 | 3.189 |
| | 16 | 17 | 1473.2 | 2.974 |
| Opt-V(No global communication) | 4 | 34 | 1444.6 | 3.032 |

6) **The term "manual vectorisation" is used many times throughout the paper. This is very misleading. Manual vectorisation would be in my opinion if the code was rewritten with avx512 vector intrinsics in C! The vectorisation used in this paper still relies on the compiler.**

**Reply:** Thanks for the comment. We agreed with the reviewer. To avoid misleading the readers, we consider that "vectorisation with compiler's directives" may be a more accurate expression. We have updated this expression throughout the revised manuscript as follows:

In Sec. 3.2, Paragraph 1, Line 2:

**"…manual strengthening of the vectorisation with the help of compiler directives…"**

In Sec.3.2, Paragraph 3, Line 1:

**"Manual strengthening of the vectorisation with compiler directives…"**

In Sec.3.2, Paragraph 3, Line 7:

**"…manually adding vectorisation directives and reconstructing the loops are needed with the help of the Intel Advisor tool mentioned above…"**

In Sec.3.2.2, Paragraph 2, Line 1:

**"In our study, strengthening of the vectorisation by adding directives and constructing loops and multithreading were done in the emission section."**

In Sec.3.2.3, Paragraph 3, Line 4:

**"…strengthening of the vectorisation was conducted…"**

In Sec.3.2.3, Paragraph 4, Line 4:

**"The strengthening of the vectorisation in the IntegrateChemistry subroutine was realised through three aspects: 1) giving the directives for the loops to instruct the compiler to vectorise the codes, including declaring no dependencies and aligning the data for efficient data accesses; 2) converting the scalar structure into vectorisation structure codes, as shown in step 3 in Figure 3, but using more complex parameter arrays to build the loop structure…"**

In Sec.3.2.4, Paragraph 1, Line 1:

**"Strengthening of the vectorisation and updating of the global communication were used in the optimisation of the diffusion module."**

In Sec.5. Paragraph 1, Line 3:

**"Strengthening the vectorisation by constructing loops and using compiler directives in GNAQPMS to make full use of the 512-bit-wide VPU on the KNL platform…"**

7) **As mentioned before the paper does not investigate what the real bottleneck of the code is and how far the code is from optimal performance. I highly recommend to create a theoretical and measured roofline model. This would allow to answer how much potential for further optimization should still be possible.**

**Reply:** Thanks for the precious comments. As we discussed in the Question 2 above, it is not easy to investigate the roofline of every section in the model. We have provided the HPC-performance reports of the Base-V and Opt-V models in Figure R1 and R2. According to the testing results, we consider that the model still has large potential for optimization. And our goal is that the model can get the speed of 5-10 model-year/day with global $0.25°x0.25°$ horizontal resolution which is suitable for high resolution long-term climate change simulation.

8) **I would love to know how many of all floating point operations are vectorized. I understand that counting floating point operations on Intel architectures through hardware counters is not very accurate. Maybe you could give some estimates from what the vectorisation report shows you?**

**Reply:** Thanks for your kind suggestion. The measuring of float point could be done by Vtune, and as the reviewer mentioned, it remains the errors of GFLOPS. However, from Figure R1 and Figure R2, we could find that the GFPLOPS increases from 93.74 to 279.479, the FPU utilization increases from 2.9% to 9.6% after optimization, and the vector capacity usage is increasing from 24% to 65%, which illustrate the improvement of code efficiency.

**Response to the General Comments of Referee #2:**

1) **"The manuscript presents results of efforts to improve the performance and scalability of a global atmospheric chemistry model, the QNAQPMS, using the Knights Landing version of the Intel Xeon Phi. Gains are relative to the original unoptimized code on both KNL and on Broadwell, Intel's latest-generation multi-core Xeon processor. The subject of the work, increasing simulation capability of QNAQPMS using next generation processors to improve computational performance and scaling, represents a noteworthy contribution to modeling science within the scope of Geoscientific Model Development. The contribution comes in the form of new concepts (next generation processors) and methods (measuring performance, identification of bottlenecks, and optimization through restructuring of application code). The approach and methods appear to be valid and reasonable, and the references to related work are appropriate and sufficient."**

**Reply:** Thanks for the praise of the reviewer. We believe this work could be a good example for other model developers who want to transplant their models to KNL platform.

2) **"This reviewer did not evaluate the manuscript with respect to scientific reproducibility. However, the verification of model output from the optimized version appears to be based on merely eyeball comparisons of color contour plots. The lack of a suitable objective verification technique would make it impossible to be sure the results were reproduced."**

**Reply:** Thanks for your precious comments. Actually, we designed two mechanisms to verify our results from the two model versions (Opt-V, Base-V). At first, we added an extra module into GNAQPMS to test our results. The function of this module is to output the concentration of specific chemical species after each chemical and physical processes. The MPI processes write their own data into their own files respectively at the same time, and each chemical and physical modules would only run one time to insulate the effect of other modules. Then, an additional program reads the files from the two version of GNAQPMS and calculates the relative and absolute errors. This method is similar with sampling the results during the running period. By this way, we can find the major errors due to every sections. On the other hand, we plotted the spatial distribution of modeling

results from the two models after a long time integration, as we did in the manuscript. Actually, we have calculated the difference of the two plots in Figure 5 (in the manuscript). But considering the beauty, the spatial distribution of the difference between the two modeling results were not provided. According to the first step, after a series of test and debugging, we found that the compile flag, such as -xCore-AVX2 and –xMIC-AVX512, would affect the results because of the sensitiveness of calculation accuracy. The –fp-model flag could reduce the error raised by advection, but the error could not be completely diminished because of numerical sensitivity.

Generally, there is no obvious difference between the spatial distribution by calculating the Relative Errors in Figure R3 and Relative Mean Square Errors in Figure R4. The REs of almost all grids are lower than 1%, and we think this is acceptable.

In the revised manuscript, we have replaced Figure 5 with Figure R3 in this document to show the differences between the two modeling results, and the following Paragraph has also been added in Sec. 4.2:

**"By comparing the model output results and plotting the spatial distribution images with the relative error (RE) shown in Figure 5, we can see, in the third column, that the RE was small (<1%) enough. The optimisation does not introduce an "erroneous" concentration for any atmospheric specie, and, therefore, it is reliable. However, the error could not be completely diminished because of the numerical sensitivity of the advection algorithm."**

3) **"Regarding the presentation of the results, the clarity and the level of detail are very poor. There are significant gaps in the explanation of, for example, the analysis of hotspots and bottlenecks in the code, and their relative impacts on performance. The use of VTune is mentioned, but how do the authors determine whether a particular hotspot is performing inadequately, and if so, what is the specific bottleneck? Insufficient use of vectors? Hardware threads? Memory bandwidth or latency? Load imbalance?"**

**Reply:** Thanks for the comments. The reviewer required to present more details about the processes of identifying the bottlenecks and optimization methods. We agreed with the reviewer and we have done the relevant modification in the revised manuscript.

Overall, three steps were used to complete the modernization of model codes: 1) Performance profiling; 2) Single node optimization; 3) Multi nodes optimization.

The bottleneck identification in this study is based on several common ways such as the Vtune tools, the application behavior analysis and characteristic extraction. The Intel Vtune Amplifier (https://software.intel.com/en-us/intel-vtune-amplifier-xe/), Intel Advisor (https://software.intel.com/en-us/intel-advisor-xe) and Intel Trace Analyzer and Collector (ITAC, https://software.intel.com/en-us/intel-trace-analyzer) can identify most of the bottlenecks of the model codes. The ITAC is mainly used to analyze the MPI performance and communication problems, including the time consuming of MPI functions and OMP threads. Figure R5 shows the graphic report of the load balance situation of GNAQPMS (using 72 MPI processes), which comes from the ITAC. The load balance situation of each MPI process is displayed by pie figure, and it illustrates that load balance is not the main problem of GNAQPMS.

[Figure]

**Figure R5. Load balance situation of GANQPMS on single CPU node detected by ITAC.**

Vtune tools currently could only be used on single node. However, it contains more functions to help us to identify the bottlenecks. The reviewer mentioned the bandwidth problem, and Vtune could collect the relevant information as well as finish the visualization as showed in Figure R6.

[Figure]

**Figure R6. The bandwidth situation during the model running detected by Vtune.**

Vtune can detect the hotspot functions of the model codes. Figure R7 presents the hotspots in the Base-V GNAQPMS detected by Vtune. The red bar indicates low CPU usage efficiency. The list of hotspots would clarify the priority of optimization work. Moreover, Figure R8 shows the hotspots in the Opt-V GNAQPMS with better CPU usage efficiency. Optimization of model codes depends on the analysis of hotspots. The Intel Advisor could help to do the vectorisation and multi-threads work. And the bandwidth issues could be detected by Vtune through analyzing the memory-access.

[Figure]

**Figure R7. Hotspot functions detected by Vtune in the Base-V GNAQPMS.**

[Figure]

**Figure R8. Hotspot functions detected by Vtune in the Opt-V GNAQPMS**

In the revised manuscript, we added a general description of how to detect the bottlenecks of the model codes in Sec. 3.1 as follows:

**"To analyse the insight performance bottleneck of GNAQPMS, we used the Intel VTune Amplifier (https://software.intel.com/en-us/intel-vtune-amplifier-xe/), Intel Advisor (https://software.intel.com/en-us/intel-advisor-xe) and Intel Trace Analyzer and Collector (ITAC; https://software.intel.com/en-us/intel-trace-analyzer). The VTune tools can do the analysis of the performance in high-performance computing (HPC), memory access, thread profiling with locks and waits analysis, floating-point operations per second (FLOPS) and floating point unit (FPU) utilisation analysis and detection of hotspot functions. By using the VTune HPC performance detection tool, we could report the general performance, e.g. GFLOPS, bandwidth, CPU and FPU utilisation, through a simple report. Table 1 presents the general indicators detected by the VTune HPC performance detection tool for the two models. Moreover, an obvious increase in GFLOPS was detected from 93.741 to 279.479. The Memory Bound in Table 1 indicates the fraction of slots where a pipeline could be stalled owing to the demand load or store instruction, and the values of 9.2% of Base-V and 12.7% of Opt-V indicate that the bandwidth is not the limitation of our model. The FPU utilisation was also improved from 2.9% to 9.6%, although there is still room for improvement. Further analysis of the hotspots and bandwidth could be detected by Hotspot and Memory-Access in VTune, respectively. Hotspots are the segment codes that consume most of the time during the running of the model. Therefore, optimising these hotspot parts will be more helpful to improve the**

**speed and efficiency of the model codes, and Figure S1 in the supplement shows the hotspots in Base-V GNAQPMS with low CPU utilisation. Moreover, using the Intel Advisor tool could help to learn about the vectorisation and bandwidth situation of the hotspot functions and modules. The Intel Advisor tool could also provide some information about the limitation of vectorisation or the reasons why vectorisation cannot be realised, as well as the primary solutions to the users to do the full vectorisation work. Furthermore, the realisation of multi-threads could be done with the help of the Intel Advisor tool. As for the MPI performance, ITAC could provide the parallel MPI balance information and communication profiling, which is auto-visualised by ITAC to analyse the MPI performance, as shown in Figure S2. However, the more significant step is designing the corresponding solutions for the hotspots or bottlenecks with the help of the tools mentioned above. Moreover, timely test and validation should be done after the optimisation. This whole process, as mentioned in Sec. 1, could be repeated many times to try different alternatives and gain a satisfactory performance."**

At the same time, the Figure R5, R7 and R8 in this response were added to the supplementary material as Figure S2, S1 and S3.

**4) "The use of OpenMP threading is also insufficiently motivated or explained."**

Reply: Thanks for the comments. As to OpenMP, our concerns are mainly about the following two aspects. On the one hand, our goal is to accelerate the model and improve the scalability, and our primary desire is to replace the MPI processes by the relative cheap OpenMP threads as many as possible on each calculation node, which is adopted in other large-scale models, such as CESM. On the other hand, considering the features of multi-core and low frequency of KNL, large number of pure MPI processes would lead to expensive communication cost for KNL and it is inevitable to do the hybrid optimization.

In the revised manuscript, the following description is added to Sec. 3.2, Paragraph 1, Line 6:

**"KNL contains many low-frequency cores, and each core contains four hyper-threads. Considering the relatively expensive overhead of communication in MPI for many cores of KNL, we adopted the hybrid parallel mode by using OpenMP and MPI. Furthermore, the OpenMP threads could fully use the hyper-threads in KNL, and the cheap communication cost of OpenMP could help to improve the scalability."**

5) **"This reviewer also takes issue with including "changing global communication from interface-files writing/reading to MPI functions" among the list of five optimizations discussed, since this change appears to address a fundamental deficiency in the parallel design of the original application, not one specific to optimizing for the Intel Xeon Phi processor. This reviewer suggests removing discussion of this optimization from the paper and using the space to more fully explain the other four optimizations. If, however, the authors wish to include discussion and results of the global communication optimization in this paper, its effect on performance and scaling should be clearly separated from the other optimization results. As currently presented, the reader cannot determine what effect this has relative to the effects of the KNL-specific optimizations."**

**Reply:** Thanks for the comments. We should declare that none of the optimizations adopted in this paper is specific for the KNL platform. Moreover, performance portability is one of the advantages of KNL. Once the code modernization work is done for KNL, the performance can be easily ported to new generation Intel CPU (e.g. Broadwell, Skylake) sharing hardware features with the same code. As mentioned by the reviewer, the interface-files could be a problem for any parallel architecture, but this problem is part of our work to deal with. Moreover, this paper describe the common optimization measures as well as the special optimization measures for our model. We have separated the effect of global communication optimization on model performance. Since the overhead of file I/O increases much faster than that of collective MPI functions as the number of nodes increases, this optimization could be important to the model scalability. However, the Endeavor cluster has updated its nodes configuration, and the benchmark CPU (E-5 2697 V4) nodes have been replaced. Therefore, we could not provide the scalability effects based on the previous testing results. We retested the single node results and showed in Table R1, including the results without the global communication. The results show that the optimal speedup on CPU and KNL are 2.77 and 3.50, respectively. Without the global communication optimization, the speedups drop to 2.69 (CPU) and 3.03 (KNL), respectively (Table R1). Generally, global communication optimization could improve the salability and the performance especially those of KNL.

In the revised manuscript, we added **Table R1** in this document as **Table 4**, and removed the previous Figure 6. The corresponding description was also added in Sec. 4.3 as follows:

"Different combinations of OpenMP and MPI processes were tested on a single node, and the results are shown in Table 4. The speedup of the best combination for Opt-V GNAQPMS reached 3.51× on KNL and 2.77× on the CPU, compared to that of Base-V GNAQPMS, and the KNL platform had an advantageous speedup of 1.26× over the CPU platform. At the same time, without the global communication optimisation, the speedup of these combinations was 3.03× on KNL and 2.70× on the CPU. In addition, these results indicated that KNL was affected more than the CPU since the KNL cores have a lower frequency for the I/O."

6) **"Because of these problems, which go beyond language and grammatical issues, this reviewer recommends the paper be rejected and reconsidered only after significant revision."**

**Reply:** Thanks for your comments. We believe this work could be a good example for other model developers who want to transplant their models to KNL platform. And we have done additional testing and analysis and revised the manuscript according to the reviewer's comments and suggestions. As for the language and grammatical issues, we have invited the English language service company **Elsevier** to do the copy editing for English. The certificate is presented in the response to the general comments of referee #1.

**Response to the Specific Comments of Referee #2:**

7) **Pages 1 and 2, Introduction. Discussion of impacts is incomplete. What levels of performance does QNAQPMS currently provide for simulation science using the model? What scientific problems are currently possible? What scientific problems are beyond reach with current QNAQPMS performance and scaling. How much more performance. Be specific.**

**Reply:** Thanks for the comments. GNAQPMS is designed for global atmospheric aerosol and chemistry simulation. Its applications include temporal and spatial evolution of atmospheric composition (ozone, black carbon, sulphate, nitrate, dust, seasalt et al.), providing boundary conditions for regional models, intercontinental long-range transport, long-term climate change (aerosol-cloud-radiation interaction), and it also acts as a key component of the Earth System Model of Chinese Academy of Sciences (CAS-ESM). Currently, GNAQPMS can only run on CPU

platform, and its parallel scalability and computation speed are about 8 CPU nodes and 46 hours per model year at 1°x1° resolution (excluding model I/O). This model performance is suitable for short-term or medium-term (5 years or less) simulation but not suitable for long-term simulation (30 years or more, needs more than 2 months of computation time). Besides, the model computation time will further increase when model I/O is included or higher model resolution (e.g. 0.25°x0.25°) is used. Therefore, it cannot be directly coupled into earth system model and used for long-term climate change simulation. By optimization of model codes and usage of new hardware, we aim to greatly improve the model parallel scalability and computation speed. The target computation speed in the future is about 5-10 model years per day (including model I/O) at 0.25°x0.25° resolution. This is an ambitious goal and needs a lot of hard work. We plan to improve the parallel computation speed of GNAQPMS in the first step, and improve the model I/O efficiency in the next step.

**In the revised manuscript, the above description about the current model level and the model optimization goal was added to Sec. 1 Paragraph 4 to make the impact and purpose of this study more specific.**

8) **Page 3, Section 2.1, Model Description of GNAQPMS. (Suggestion) What is the difference between GNAQPMS and NAQMPS? Basing the description of GNAQPMS on the NAQPMS presupposes knowledge about the NAQPMS. There is already a reference to NAQPMS but also add a few sentences of background on the NAQPMS. Explain why the authors focus GNAQPMS and not at the NAQPMS.**

**Reply**: Thanks for the suggestions. The GNAQPMS model is a global multi-scale chemical transport model based on the Nested Air Quality Prediction Modeling System (NAQPMS) (Wang et al., 2006), developed at the Institute of Atmospheric Physics, Chinese Academy of Sciences. NAQPMS is a 3-D regional Eulerian model which has been rigorously evaluated and widely applied to simulate the chemical evolution and transport of ozone (Li et al., 2007; Tang et al., 2010), the distribution and evolution of aerosol and acid rain over East Asia (Wang et al., 2002; Li et al., 2011; Li et al., 2012) and to provide operational air quality forecasts in mega cities such as Beijing, Shanghai and Guangzhou (Wang et al., 2010; Wu et al., 2012; Wang et al., 2009). GNAQPMS and NAQPMS use the similar model framework, physical and chemical parameterization schemes and parallel computation techniques. The optimization achievements in this study could be largely

shared between these two models. NAQPMS is mainly used for regional high resolution air pollution simulation and routine air quality prediction. The typical time scale of these applications is several days or months. And the current computation performance of NAQPMS can generally meet the demand of these applications. GNAQPMS is designed for global-scale, long-term atmospheric aerosol and chemistry simulation, and it is also online coupled to the Earth System Model of Chinese Academy of Sciences (CAS-ESM) for study of climate change. These applications have typical time scale of more than 10 years, and raise very high requirements for model computation speed (e.g. 5-10 model years per day). Obviously, there is a large gap between the current computation performance of GNAQPMS and the actual need. Therefore, we choose GNAQPMS to start the code optimization.

**In the revised manuscript, a brief description of NAQPMS as shown above was added to Section 2.1 Paragraph 1, and the reason why we focus on GNAQPMS is also given.**

9) **Page 4, line 3: "Since the memory processor is not dominant". Unclear, explain further: what do the authors mean specifically by "memory pressure". Memory working set size? Memory bandwidth requirement of this application. How have they determined it is not dominant?**

**Reply:** Thanks for the comments. The memory bandwidth profiling given by VTune shows that the memory bandwidth requirement of the model is far below the peak capability of both CPU and KNL platforms, as shown in Figure R6. In addition, the memory footprint of this workload is about 3G, which can be fully accommodated by the 16GB MCDRAM on KNL.

In order to explain clearly, this sentence has been modified in Sec. 2.2, Paragraph 1, Line 8:

**"Since GNAQPMS is not limited by the memory working set size detected by the VTune Memory-Access tool, the cache mode was chosen in our experiment."**

10) **Page 4, line 9: Spell out first use of TLS to represent thread local storage. It is spelled out in the abstract, but needs to be spelled out again in main body of paper.**

**Reply:** Thanks for the suggestion. TLS has been spelled out in the revised manuscript as following in Sec. 3, Paragraph 1, Line 2:

**"…reducing unnecessary memory access, reducing the thread local storage (TLS) and**

**changing the way the global communication works in GNAQPMS."**

**11) Page 5, Paragraph beginning line 6 and all of Section 3.2.1. As noted in the general comments above, the use of files for global communication instead of MPI reductions and gathers is problematic for any modern parallel architecture, not just the KNL. Strongly recommend removing discussion of this optimization, including the entire Section 3.2.1, from the paper. It is relevant to optimizing for KNL.**

**Reply:** Thanks for the precious suggestion. This paper describe the common optimization measures as well as the special optimization measures for our model, and low frequency cores on KNL could lead to a fact that I/O could have more effect on KNL compared with that on CPU (Table R1) To keep the integrity of our work, we kept the discussion in Sec. 3.2.1 and separated the contribution of global communication optimization.

Because of the adjustment of the platform for single node testing, we retested the single node results and showed in Table R1, including the results without the global communication. The results show that the optimal speedup on CPU and KNL are 2.77 and 3.50, respectively. Without the global communication optimization, the speedups drop to 2.69 (CPU) and 3.03 (KNL), respectively (Table R1). Generally, global communication optimization could improve the salability and the performance especially those of KNL.

In the revised manuscript, we added **Table R1** in this document as **Table 4**, and removed the previous Figure 6. The corresponding description was also added in Sec. 4.3 as follows:

  **"Different combinations of OpenMP and MPI processes were tested on a single node, and the results are shown in Table 4. The speedup of the best combination for Opt-V GNAQPMS reached 3.51× on KNL and 2.77× on the CPU, compared to that of Base-V GNAQPMS, and the KNL platform had an advantageous speedup of 1.26× over the CPU platform. At the same time, without the global communication optimisation, the speedup of these combinations was 3.03× on KNL and 2.70× on the CPU. In addition, these results indicated that KNL was affected more than the CPU since the KNL cores have a lower frequency for the I/O."**

**12) Page 6, line 11. "Cyclic order"? Do the authors mean loop nesting order?**

**Reply:** Thanks for the comments. Yes, it means the loop nesting order. The sentence was revised in

Sec. 3.2.2, Paragraph 2, Line 2 as follows:

**"we changed the nesting order of loops from j, i, igas to igas, j, i,…"**

13) **Page 6, line 12. "We cancelled the calling of the subroutine ... and made it an internal function in main program." This technique is referred to as inlining. Did the authors look at inlining reports from the Intel compiler to see if this could have been accomplished automatically or with the help of directives and compiler options, without manually restructuring the code? Also, the authors mean to say the "calling subroutine", not the "main program".**

Reply: Thanks for the comments. Yes, we've tried the "-ipo" option for automatic inlining and checked the compiler report. The report on this subroutine said "Inlining would exceed -inline-max-size value", which means inlining could not be done by the compiler due to unsatisfied heuristic. Therefore, we have to manually restructure the code at this step, and the calling site happens to be in the main program. According to the reviewer's comments, we modified the sentence in Sec. 3.2.2, Paragraph 2, Line 4 as follows:

**"Second, since the subroutine get_ratio_emit() is too big to be inlined automatically by the Intel compiler, we manually inlined it in the calling site of the main program to improve the calling efficiency and facilitate the vectorisation."**

14) **Page 6, line 14. "... using parameters to convert scalar structure to vector structure." Unclear what "coverting scalar to vector" means, nor does it appear from step 3 Figure 3 that this is what is going on.**

Reply: Thanks for the comments. As the reviewer mentioned, this part means the step 3 in Figure 3 in the manuscript. In this part, we used the parameter arrays to construct the loop to do vectorisation calculation. Therefore, we modified this sentence in Sec. 3.2.2, Paragraph 2, Line 6 to explain it more clearly:

**"Third, vectorisation was involved in the emission section in the model by using the parameters to convert a scalar structure of assignment value to variables to a vector structure, which is shown in step 3 in Figure 3."**

**15) Page 6, line 15. "... we added the directives, clauses, declarations and syntax comment of OpenMP outside the outermost loop as shown in box 4." This sentence appears to be the only discussion in the paper of how and why OpenMP threading was added to the code. Given that this is one of the five optimizations being presented, this is insufficient. It's not clear why the authors felt it was important to add OpenMP threading nor whether, from the results, it provided a benefit.**

**Reply:** Thanks for the comments. As described in the response to the general comments at the beginning (Question 4), our concerns are mainly about two aspects. On the one hand, our goal is to accelerate the model and improve the scalability, and our primary desire is to replace the MPI processes by the relatively cheap OpenMP threads as many as possible on each calculation node, which is also adopted by other large-scale models, such as CESM. On the other hand, considering the features of multi-core and low core frequency of KNL, large number of pure MPI processes would lead to expensive communication cost on KNL and it is inevitable to do the hybrid optimization. Currently, our OpenMP optimization did not achieve the primary goals, and the optimal number of threads on KNL and CPU are 4 and 6 (Table R1), respectively. As mentioned in the manuscript, the results we described are our first move and the OpenMP optimization is relatively simple. In the future, we will do more investigation into this part and rebuild the OpenMP code structure.

In the revised manuscript, the following description about OpenMP is added to Sec. 3.2, Paragraph 1, Line 6:

**"KNL contains many low-frequency cores, and each core contains four hyper-threads. Considering the relatively expensive overhead of communication in MPI for many cores of KNL, we adopted the hybrid parallel mode by using OpenMP and MPI. Furthermore, the OpenMP threads could fully use the hyper-threads in KNL, and the cheap communication cost of OpenMP could help to improve the scalability."**

**16) Page 7, line 15. "2) updating some code segments to the the serial codes to construct vectorisation." What does this mean?**

**Reply:** Thanks for the comments. It refers to construct loop to do the vectorisation calculation, which is similar with the step 3 in Figure 3 in the manuscript. In order to avoid the confusion of this

sentence, the original sentence in Sec. 3.2.3, Paragraph 4, Line 6 is modified as follows:

**"…converting the scalar structure into vectorisation structure codes, as shown in step 3 in Figure 3, but using more complex parameter arrays to build the loop structure…"**

17) **Page 7, lines 17-25. A reader might make a number of educated guesses about what the authors are saying was the bottleneck and how it was fixed, but it is not clear at all from the text. What codes were "added by the compiler?" Was the issue a copyin/copy-out problem for thread-local variables that were listed as private/firstprivate in the OpenMP directives? If so, how did adding the CBMZOBJ argument fix this? By passing-by-reference? If so, how were data races avoided? Were they avoided because the CBMZOBJ objects themselves were THREADPRIVATE? There's a lot of important detail missing from this discussion.**

**Reply:** Thanks for the precious comments. Firstly, please allow us to clarify the performance issue with common variables in OpenMP TLS. The TLS is introduced for variables in named common blocks when using thread private OpenMP directive, and allocated for each thread on thread creation. These variables are private to each thread and global within the thread. When a thread references a common variable in its TLS, the memory address of TLS is first located by calling an OpenMP library function with the thread ID, then the common variable is addressed within the TLS space. Even for the references to common variables within the same named common block in the same subroutine, the above process is repeated for every variable, rather than addressing the TLS and common block only once. Since calling the OpenMP library function for TLS addressing is expensive, and there are many references to these common variables in the user subroutines, the total overhead of using common variables in TLS is extremely high. Linking against static OpenMP library can alleviate the calling cost partially but the cost is still unbearable.

Basically, our solution in this study is to construct a derived type (CBMZTYPE) object to eliminate the expensive OpenMP function calling. In the optimized code, the common variables are removed from named common blocks and added to the CBMZTYPE as its members. Using PIRVATE list in the OpenMP directive, each thread owns a private copy of object instance of CBMZTYPE, i.e. the cbmzobj variable. Since the cbmzobj is located on thread local stack, the references to its member variables require only simple relative addressing on stack, with simple yet efficient

instructions. Meanwhile, since the common variables in the original code are no longer global and visible within the user subroutines, a formal parameter (argument) of cbmzobj is added to the subroutines using the variables. The additional overhead of passing the address of cbmzobj to the subroutine is quite small. Therefore, the cost of referencing common variables in TLS is greatly reduced with the derived type object. The codes were "added by the compiler" means the codes of referencing the common variable in TLS, including the calling to OpenMP library function for TLS address, and then accessing the common variable within the TLS space. And it is a reference cost issue, for variables in named common blocks that are listed in the THREADPRIVATE directive. And this issue is not fixed by adding the CBMZOBJ argument, but by constructing the CBMZOBJ object to store the variables in the original common blocks. Since the CBMZOBJ object is thread private and is on the thread local stack, the references to its members are quite cost efficient.

In the revised manuscript, the description of TLS optimization was added in Sec. 3.2.3, Paragraph 4, Line 14 as follows:

**"Using a PRIVATE list in the OpenMP directive allows each thread to own a private copy of the object instance of cbmztype, i.e. the cbmzobj variable. Since cbmzobj is located on a thread local stack, the references to its member variables require only simple relative addressing on the stack, with simple yet efficient instructions. Meanwhile, since the common variables in the original code were no longer global and are now visible within the user subroutines, a formal parameter (argument) of cbmzobj was added to the subroutines using the variables. The additional overhead of passing the address of cbmzobj to the subroutine is quite small. Therefore, the cost of referencing common variables in the TLS is greatly reduced with the derived type object."**

**18) Page 7, line 32. "...which spent 10 percentages and 8 percentages." One would prefer to see actual timings instead of percentage of runtime in discussion of performance improvement.**

Reply: Thanks for the comments. Firstly, we updated the data in Fig. 2 because the previous results came from an old testing. And the description of Fig.2 in the manuscript was revised accordingly. The aim of this part is to emphasize the role of different sections of the time consumption. The impact of the optimization on these sections is showed in Sec. 3.2.4, Paragraph 1, Line 1, with the

speedup. According to the comments of the reviewer, we removed the old Paragraph 1 in Sec. 3.2.4 and added the walltime as well as the speedup to illustrate the improvement:

**"According to the performance on the single node, the diffusion module could get a 1.99× speedup (from 241.97 s to 121.19 s) on the CPU platform and a 3.31× speedup (from 241.97 s to 73.05 s) on the KNL platform."**

**"Finally, the optimised wet deposition module got a 5.92× speedup (from 498.01 s to 84.13 s) on the KNL platform, which was much higher than the 3.19× speedup (from 498.01 s to 156.13 s) on the CPU platform."**

19) **Page 7, line 36. "1.78 speedup on the CPU platform and 2.39 speedup on the KNL." How much of these speedups were from the manual vectorisation and how much from the optimization of global communication? (Again, this reviewer suggests not considering global communication optimization at all, but if it is discussed, show effects separately from other optimizations).**

**Reply:** Thanks for the comments. As mentioned in the response for Question 5 and 11 above, we tend to discuss the optimization of global communication as part of our work, and we have separated the contribution of this part from the total speedup as required by the reviewer.

Because of the adjustment of the platform for single node testing, we retested the single node results and showed them in Table R1, including the results without the global communication optimization. The results show that the optimal speedup on CPU and KNL are 2.77 and 3.50, respectively. Without the global communication optimization, the speedups drop to 2.69 (CPU) and 3.03 (KNL), respectively (Table R1).

In the revised manuscript, we added **Table R1** as **Table 4**, and removed the previous Figure 6. The corresponding description was also added in Sec. 4.3 as follows:

**"Different combinations of OpenMP and MPI processes were tested on a single node, and the results are shown in Table 4. The speedup of the best combination for Opt-V GNAQPMS reached 3.51× on KNL and 2.77× on the CPU, compared to that of Base-V GNAQPMS, and the KNL platform had an advantageous speedup of 1.26× over the CPU platform. At the same time, without the global communication optimisation, the speedup of these combinations was 3.03× on KNL and 2.70× on the CPU. In addition, these results indicated that KNL was**

**affected more than the CPU since the KNL cores have a lower frequency for the I/O"**

20) **Page 8, lines 7-10. "The horizontal resolution of the model is 1°×1°, which indicates that the modelling domain contains 360×180 grids. And the number of vertical layers is 20, while the time step for integration is 600 seconds in the test case. The test case was designed to test the performance of GNAQPMS on single node of CPU and KNL platform, and multi-nodes on different platform clusters." This workload is very small and probably not suitable for KNL clusters, which require a high degree of parallelism to be efficient. (The authors make note of this on page 9, lines 32-33). The paper describe a representative workload for scientific simulations using the GNAQPMS and provide some discussion of how the performance results presented are relevant to an actual scientific workload.**

**Reply:** Thanks for the comments. We generally agree with the reviewer. Currently, due to limited HPC resources and poor model computation performance, the typical resolution for global aerosol and chemistry simulation in scientific research is about 2°x2°. It makes sure that a several year's model simulation can be accomplished in a month. The test case in this study was chosen based on the following reasons: 1) The configuration with 1°x1° horizontal resolution and 600 s integration time step is common in scientific applications. This model configuration has also been used in several previous studies (Chen et al., 2015). By using this configuration, the achievements of this study can be directly applied to actual scientific applications. 2) The test case is a medium-scale workload for global chemistry simulation, which allows us to carry out a lot of debugging and testing. 3) This is the first time to port and optimize GNAQPMS on the KNL platform. A lot of fundamental work is needed to optimize the code and solve potential bugs. Therefore, choosing a medium-scale test case would be a good start.

We agree that a large workload with high model resolution is probably more suitable for KNL clusters, and it might get better parallel scalability. In addition, based on our tests, the bottleneck of MPI global communication and fragmental OpenMP parallel regions are also main reasons for the poor parallel scalability of GNAQPMS on KNL clusters. Global high resolution simulation is a clear trend for the development of chemical transport models. Testing and optimizing GNAQPMS with a super large workload (e.g. 0.25°x0.25° or 0.1°x0.1°) will be the next emphasis in the near future. In the revised manuscript, further description of the test case is added to Sec. 4, Paragraph1, Line 6

as follows:

**"This test case was an actual scientific workload and had a medium scale of calculation amount; therefore, it allowed us to carry out much debugging and testing within a short time."**

21) **Page 8, line 15. "The Intel Corporation provides the High Performance Computing environment for the test." Was this Intel's Endeavor cluster? Perhaps mention this as well.**

**Reply:** Thanks for the comments. There are two sets of platforms used in this study. The single node tests were conducted on **Cthor Lab.** of Intel Corporation and the cluster tests were done on **Intel Endeavor** cluster. Considering the stability of the test environment, we adopted the machines in **Cthor Lab.** for the single-node test. According to the suggestion, we have mentioned the names of the test environment in the revised manuscript in Sec. 4.1, Paragraph 1, Line 1:

**"The Cthor Lab. of Intel Corporation was adopted for the single-node tests owing to its relatively steady environment, and Intel's Endeavor cluster was used for the cluster tests."**

22) **Page 8, line 20. Opt-V GNAQPMS has been compiled on CPU and KNL platform, respectively." Not sure what this means. Are the authors actually compiling the codes on the respective platforms? If so, why? Is there some difference expected between a native-compiled and a cross-compiled executable?**

**Reply:** Thanks for the comments. Both native and cross compiling work, and there are no differences between them. Actually, we compiled the same codes with different compiler flags as showed in Table 2 on the same CPU platform. In order to avoid such confusion, we modified this sentence as follows in Sec. 4.1, Paragraph 1, Line 8:

**"…and Opt-V GNAQPMS was compiled for the CPU and KNL platforms with its own compile flags, shown in Table 3."**

23) **Page 8, line 21. "... the -xCore-AVX2 and –xMIC-AVX512 compile flags were not used for the advection module..." This is troubling and bears further discussion. What did the authors see that caused them to avoid these compiler flags for the advection module? Why is advection susceptible to differences but not other parts of the code such as the ODE solver?**

**Reply:** According to this comment, the answers are similar with the response to Question 3 of the referee #1. Actually, we designed two mechanisms to verify our results from the two model versions (Opt-V, Base-V). At first, we added an extra module into GNAQPMS to test our results. The function of this module is to output the concentration of specific chemical species after each chemical or physical processes. The MPI processes write their own data into their own files respectively at the same time, and each chemical and physical modules would only run one time to insulate the effect of other modules. Then, an additional program reads the files from the two versions of GNAQPMS and calculates the relative errors and absolute errors. This method is similar with sampling the results during the running period. By this way, we can find the major errors due to every sections. On the other hand, we plotted the spatial distribution of modeling results from the two models after a long time integration, as we did in the manuscript. Actually, we have calculated the difference of the two plots in Figure 5 (in the manuscript). But considering the beauty, the spatial distribution of the difference between the two modeling results were not provided. According to the first step, after a series of test and debugging, we found that the compile flag, such as -xCore-AVX2 and –xMIC-AVX512, would affect the results because of the sensitiveness of calculation accuracy. The –fp-model flag could reduce the error raised by advection, but the error could not be completely diminished because of numerical sensitivity.

Generally, there is no obvious difference between the spatial distribution by calculating the Relative Errors in Figure R3 and Relative Mean Square Errors in Figure R4. The REs of almost all grids are lower than 1%, and we think this is acceptable.

In the revised manuscript, we have replaced Figure 5 with Figure R3 in this document to show the differences between the two modeling results, and the following Paragraph has also been added in Sec. 4.2:

**"By comparing the model output results and plotting the spatial distribution images with the relative error (RE) shown in Figure 5, we can see, in the third column, that the RE was small (<1%) enough. The optimisation does not introduce an "erroneous" concentration for any atmospheric specie, and, therefore, it is reliable. However, the error could not be completely diminished because of the numerical sensitivity of the advection algorithm."**

**24) Page 9, line 2. "...were confirmed to be identical." This is not a persuasive verification**

**method. An eyeball comparison of plots would not be considered sufficient to confirm that results are identical. Provide difference plots and RMS difference statistics. Page 9, line 20. "...when the computing scale is fixed." Right word? Instead maybe workload? problem size?**

**Reply:** Thanks for your kind comments. The answer for the validation of model results has been given in the response to Question 23 above.

According to the suggestion, the sentence in Sec. 4.4, Paragraph 1, Line 2 is modified to explain more suitable:

**"...when the workload is fixed".**

25) **Page 9, line 24. "After optimization, the parallel scalability of GNAQPMS is greatly improved..." This seems counterintuitive, since vectorisation and other node-performance optimizations that improve performance on each node from should make the code less well, assuming that interprocessor communication overheads are the same. It's important to distinguish here (1) what parts of the code are being timed for the scaling measurements. (2) Which optimizations are contributing to the improved scalability and which may be improving performance but working against strong scaling.**

**Reply:** Thanks for the comments. As mentioned in Section 3.2, the optimizations include the global communication optimization that replaces interface-files writing/reading with collective MPI functions. This optimization reduces the communication overhead greatly, since the overhead of file I/O increases much faster than that of collective MPI functions as the number of nodes increases. For the optimized code, since the communication improvement is much more than that of the computation part, we observe a great improvement in the parallel scalability.

To (1) question, we evaluate the run time for "the core calculation portion of the model", as mentioned in Page 9, Line 23.

To (2) question, we would declare that the global communication optimization mainly contributes to the scalability improvement, and other optimizations mainly contribute to the computation improvement. But we would like to further point out that single-node optimization does not necessarily work against strong scaling. There are at least two exceptions. One is the OpenMP parallelization that could hide the latency of MPI point-to-point communication partially or fully.

In some cases, it can help improve the single-node performance as well as the cluster scalability. Another is the CA-KSMs (Communication-Avoiding Krylov Subspace Methods) developed by Professor Demmel at Berkeley, at algorithmic level, which optimize the global communication and expose the opportunity for local computation optimization at the same time.

26) **Page 9, line 33. "... OpenMP code segments on KNL..." What is the effect of varying the number of OpenMP threads with respect to MPI tasks? That is, using the same overall number of hardware threads across the job? What is the effect of running pure MPI? Pure OpenMP?**

**Reply:** Thanks for your kind comments. Firstly, we unfortunately found that the single node test environment (Cthor Lab.) had some adjustment of the machines, which lead to the difference of the testing results. Therefore, we retested all the results and updated the data in the revised manuscript. As suggested by the reviewer, we tested the combination of different OpenMP threads and MPI processes (Table R1). All the tests are fully using the hardware threads. The results indicate that the best combination on CPU platform is 6 OpenMP with 12 MPI processors. For KNL, we used the command line "–env I_PIN_MPI_DOMAIN=N" to pin the MPI processes to specific cores. And the combination of 4 OpenMP and 34 MPI processes could get the optimal speedup (3.509, which is 3.34 in the original manuscript).

In the revised manuscript, we added **Table R1** as **Table 4**, and removed the previous Figure 6. The corresponding description was also added in Sec. 4.3 as follows:

**"Different combinations of OpenMP and MPI processes were tested on a single node, and the results are shown in Table 4. The speedup of the best combination for Opt-V GNAQPMS reached 3.51× on KNL and 2.77× on the CPU, compared to that of Base-V GNAQPMS, and the KNL platform had an advantageous speedup of 1.26× over the CPU platform. At the same time, without the global communication optimisation, the speedup of these combinations was 3.03× on KNL and 2.70× on the CPU. In addition, these results indicated that KNL was affected more than the CPU since the KNL cores have a lower frequency for the I/O"**

27) **Page 10, Conclusion section. Future work and areas for improvement are discussed. However the conclusion should also include discussion of what levels of performance and**

**scaling are needed for simulation science using GNAQPMS. The conclusion should assess whether and how well the presented work helps the GNAQPMS users achieve these goals.**

**Reply:** Thanks for the comments. We agree with the reviewer. The levels of performance and scaling needed for simulation science using GNAQPMS has been discussed in the response of Question One. For short-term or medium-term simulations (5 years or less), the model computation speed needed is about 1 model year per day (including model I/O), while it should be increased to about 5 model years per day for long-term (30 years or more) high resolution simulations (e.g. 0.25°x0.25°). Improve single node computation speed and parallel scalability is the way to achieve the above goals. As shown in the conclusion, the single node computation speed and parallel scalability of GNAQPMS were significantly improved after code optimization. The computation speed (excluding model I/O) has been improved from about 0.5 model year per day using 8 CPU nodes to about 3.7 model year per day using 30 KNL nodes and about 8 model year per day using 40 CPU nodes, respectively. Therefore, without regard to the model I/O, the optimization work in this study has made the computation performance of GNAQPMS very close to our anticipated goal. In the next step, further work will be focused on solving problems concerning OpenMP parallel regions and global communication on KNL platform and conducting analysis and optimization of model I/O.

In the revised manuscript, further discussion of the model computation performance needed now and in the future is added to Section 5, Paragraph 2, Line 9 as follows:

**"In summary, the computation speed (excluding the model I/O) was improved from about 0.5 model years per day using 8 CPU nodes to about 3.7 model years per day using 30 KNL nodes and about 8 model years per day using 40 CPU nodes, respectively. Therefore, without regard to the model I/O, the optimisation work in this study resulted in the computation performance of the GNAQPMS being very close to our anticipated goal. In the next step, further work will be focused on merging the OpenMP parallel regions.**

28) **Figure 7. Shows speedup but not performance. Suggest adding a figure that shows performance as a function of the number of nodes. Plot as simulation seconds per second of wall clock time.**

**Reply**: Thanks for your kind comments. We added the following figure that shows the performance

as a function of the number of nodes, and this figure has been added into the revised manuscript as Figure 7:

[Figure]

**Figure R9. The model performance with increasing number of nodes.**

In the revised manuscript, the description of this figure was also added to Sec. 4.4, Paragraph 2, Line 11 as follows:

**"In addition, Figure 7 shows that the simulation speed, or model second per real second, improved with increasing number of nodes. When reaching 40 CPU nodes, the CPU cluster could do the simulation of 8 model years per day, and with 30 KNL nodes, it could do the simulation of 3.7 model years per day excluding the I/O part. The optimisation work in this study has made the computation performance of GNAQPMS very close to our anticipated goal of 5-10 model years under the coarse spatial resolution."**

**References:**

Chen, H. S., Wang, Z. F., Li, J., Tang, X., Ge, B. Z., Wu, X. L., Wild, O., and Carmichael, G. R.: GNAQPMS-Hg v1.0, a global nested atmospheric mercury transport model: model description, evaluation and application to trans-boundary transport of Chinese anthropogenic emissions, Geosci. Model. Dev., 8, 2857-2876, doi:10.5194/gmd-8-2857-2015, 2015.

Li, J., Wang, Z., Akimoto, H., Gao, C., Pochanart, P., and Wang, X.: Modeling study of ozone

seasonal cycle in lower troposphere over east Asia, J. Geophys. Res.-Atmos., 112, D22s25, doi:10.1029/2006jd008209, 2007.

Li, J., Wang, Z., Wang, X., Yamaji, K., Takigawa, M., Kanaya, Y., Pochanart, P., Liu, Y., Irie, H., Hu, B., Tanimoto, H., and Akimoto, H.: Impacts of aerosols on summertime tropospheric photolysis frequencies and photochemistry over Central Eastern China, Atmos. Environ., 45, 1817-1829, doi:10.1016/j.atmosenv.2011.01.016, 2011.

Li, J., Wang, Z., Zhuang, G., Luo, G., Sun, Y., and Wang, Q.: Mixing of Asian mineral dust with anthropogenic pollutants over East Asia: a model case study of a super-duststorm in March 2010, Atmos. Chem. Phys., 12, 7591-7607, doi:10.5194/acp-12-7591-2012, 2012.

Tang, X., Wang, Z., Zhu, J., Gbaguidi, A. E., Wu, Q., Li, J., and Zhu, T.: Sensitivity of ozone to precursor emissions in urban Beijing with a Monte Carlo scheme, Atmos. Environ., 44, 3833-3842, doi:http://dx.doi.org/10.1016/j.atmosenv.2010.06.026, 2010.

Wang, Q., Fu, Q., Wang, Z., Wang, T., Liu, P., Lu, T., Duan, Y., and Huang, Y.: Application of ensemble numerical model system on the air quality forecast in Shanghai (in Chinese), Environmental Monitoring and Forewarning, 2(4), 1-6+11, 2010.

Wang, Z., Akimoto, H., and Uno, I.: Neutralization of soil aerosol and its impact on the distribution of acid rain over east Asia: Observations and model results, J. Geophys. Res.-Atmos., 107, 4389, doi:10.1029/2001jd001040, 2002.

Wang, Z., Xie, F., Wang, X., An, J., and Zhu, J.: Development and application of Nested Air Quality Prediction Modeling System (in Chinese), Chinese Journal of Atmospheric Sciences, 30(5), 778-790, 2006.

Wang, Z., Wu, Q., Gbaguidi, A., Yan, P., Zhang, W., Wang, W., and Tang, X.: Ensemble air quality multi-model forecast system for Beijing (EMS-Beijing): Model description and preliminary application (in Chinese), Journal of Nanjing University of Information Science & Technology (Natural Science Edition), 1(1), 19-26, 2009.

Wu, Q., Wang, Z., Chen, H., Zhou, W., and Wenig, M.: An evaluation of air quality modeling over the Pearl River Delta during November 2006, Meteorol. Atmos. Phys., 116, 113-132, doi:10.1007/s00703-011-0179-z, 2012.